# REASONING VIA TEST-TIME INSTANCE-LEVEL POLICY GRADIENT IN LATENT SPACE

## ABSTRACT

Large Language Models (LLMs) typically reason through explicit, step-by-step natural-language traces. Humans, however, also rely on non-linguistic, unconscious processes, such as the inspirations that emerge during the incubation period. In this work, we introduce LATENTSEEK, a novel framework designed to enhance the reasoning capabilities of LLMs through Test-Time Instance-level Policy Gradient within the model's latent space—thus complementing explicit natural-language steps. LATENTSEEK employs policy gradient optimization to iteratively refine latent representations, guided solely by a self-generated reward signal. This allows the model to adapt its reasoning trajectory dynamically on a per-instance basis. Empirical evaluations across diverse benchmarks, GSM8K, MATH-500, and AIME2024 as well as multiple LLM families (e.g., LLaMA, Qwen) demonstrate that LATENTSEEK outperforms established baselines, including Chain-of-Thought (CoT), Best-of-N (BoN) and training-based methods. Further analysis indicates that LATENTSEEK is computationally efficient, typically converging within a few optimization iterations for average-level problems. Moreover, the model's performance improves as the number of latent update iterations increases, highlighting the benefits of exploring within the latent space. These findings highlight LATENTSEEK as a lightweight and effective paradigm for improving the reasoning capabilities of LLMs without changing their parameters.

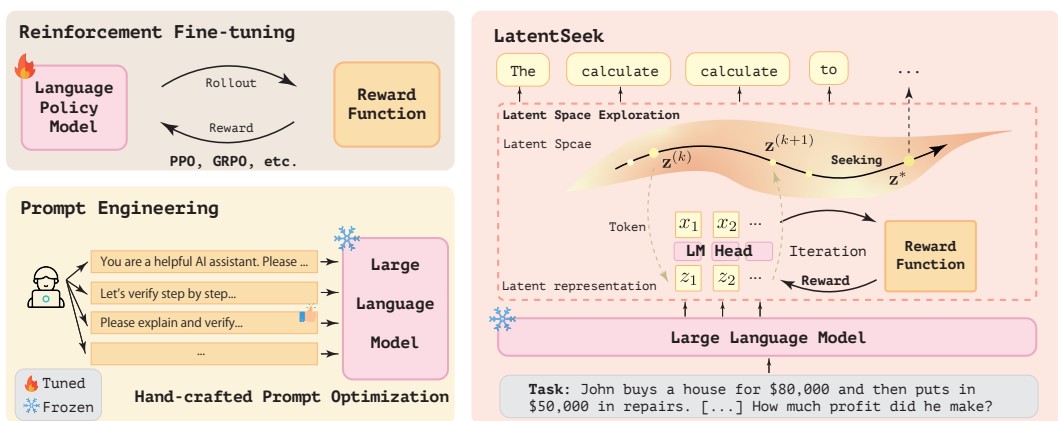

Figure 1: Comparison of LATENTSEEK with RL-based fine-tuning and Prompt Engineering. RL-based fine-tuning methods generally require iterative updates to model parameters guided by reward signals. Prompt engineering approaches depend heavily on manually designed prompts. In contrast, **LATENTSEEK** performing optimization within the latent space. Of note, the output of LATENTSEEK may be incoherent and semantically ungrounded; please refer to §3.7.

## 1 INTRODUCTION

Large Language Models (LLMs) have demonstrated remarkable reasoning abilities, largely unlocked by techniques such as Chain-of-Thought (CoT) prompting, which guides them to generate step-by-

step solutions in natural language (Achiam et al., 2023; Wei et al., 2022). This approach, however, presupposes that the optimal reasoning path for a neural network is one that mirrors human linguistic expression. However, research in human cognition suggests that effective reasoning is not confined to conscious, verbal articulation (Sio and Ormerod, 2009; Ritter and Dijksterhuis, 2014); it also involves implicit, non-verbal processes, such as the "incubation" period where inspirations can emerge without conscious effort (Sio and Ormerod, 2009; Sklar et al., 2012). These findings suggest that constraining LLMs to reason only in natural language may be unnecessarily restrictive.

An emerging line of research validates this concept, using specialized fine-tuning to "internalize" reasoning steps into sequences of continuous thought vectors (Deng et al., 2024; Hao et al., 2024). While these approaches validate the promise of latent reasoning, they rely on costly and complex training curricula to adapt the model. As illustrated in Figure 1, this aligns them with the broader paradigm of training-based methods such as supervised fine-tuning or reinforcement fine-tuning, which require iterative parameter updates that are computationally intensive and permanently alter the backbone (Ouyang et al., 2022; DeepSeek-AI, 2025). As a result, they underutilize the model's semantic capacity in its pre-trained **latent space**[1] and, in practice, present inferior performance to that of CoT. This raises a critical question: *Can we take the benefits of pre-trained latent space to perform reasoning at test time, without the need for any parameter updates?*

Motivated by these observations, we present the **first** attempt to perform seeking in the latent space by introducing **LATENTSEEK**, a framework that significantly enhances instance-level reasoning at test time. As shown in Figure 1, LATENTSEEK operates on a frozen model, circumventing the costs of fine-tuning. Unlike static **prompt engineering**, LATENTSEEK performs dynamic, iterative optimization on instance-specific latent representations that steer the pre-trained model's reasoning process without modifying its parameters. These latent representations act as a planning or control mechanism that guides the model toward better reasoning paths for each specific problem instance. We optimize latent representations at test time using the policy gradient method (Williams, 1992) to maximize reward (§2.3). Specifically, for each reasoning problem, we update the token-wise latent representations using guidance from the reward function, treating them as independent variables. In each iteration, the updated latent representations are decoded into tokens, which serve as inputs for computing the reward. Importantly, the reward function operates in a self-rewarding manner, relying solely on the model's internal capabilities without incorporating any external information. The process continues until the reward exceeds a predefined threshold or the maximum number of iterations is reached.

Our innovative latent space reasoning method is **simple yet effective**: Notably, LATENTSEEK yields average gains of 15.23 points over BoN on GSM8K, 4.72 points over Self-Reflection on MATH-500, and 6.67 points over CoT on AIME2024 (Table 1). Furthermore, when using LLaMA3.1-8B-Instruct as the backbone, LATENTSEEK surpasses prior arts including SimpleRL-Zoo (Williams, 1992) (+18.1) and Genius (Xu et al., 2025) (+12.7), on the GSM8K and MATH-500 datasets (Table 2). Further experiments ( §3.4) show that test-time performance improves with the number of update iterations, suggesting a **complementary scaling axis: the number of optimization steps in latent space**. To better characterize the latent space, we conduct idealized experiments using a perfect, ground-truth verifier that issues sharp rewards. Across all evaluation settings, this yields an average gain of 19.12 points over CoT (Table 3), **highlighting the benefits of pure exploration in the latent space**. Apart from the above, our case studies (§3.7) reveal that LATENTSEEK often attains the correct answer even when its intermediate reasoning is unintelligible to human readers. This divergence reveals that the optimal reasoning path for LLMs need not mirror human reasoning, and that **optimization in latent space may be a more native and effective paradigm**.

Our contributions can be summarized as follows:

- We introduce LATENTSEEK, an efficient yet effective method that enhances reasoning capabilities through test-time instance-level policy gradient, and demonstrate that it outperforms all baselines.
- We conduct a scaling analysis, revealing that performance at test time improves with an increased number of update iterations, highlighting the potential of test-time scaling in the latent space.
- We conduct a statistical analysis to demonstrate the efficiency of LATENTSEEK and the appropriateness of its output length. Our qualitative analysis further indicates that language models may adopt reasoning strategies that diverge from human intuition.

---

[1]In this work, we take the convention (Hao et al., 2024) that treats the transformers' output space ahead of the final language model (LM) head as latent space (Figure 1), and the vector in the space as latent representation (Figure 1); refer to §2.2 for notations.

## 2 TEST-TIME INSTANCE-LEVEL POLICY GRADIENT IN LATENT SPACE

### 2.1 PROBLEM FORMULATION: TEST-TIME INSTANCE-LEVEL REASONING

Let $\mathbf{c}$ be the context prompt of a reasoning problem instance and $\pi$ a pre-trained auto-regressive language model (LM) composed of a Transformer backbone $\pi_{\text{Transformer}}$ (Vaswani et al., 2017) and a LM head $\pi_{\text{LM-head}}$. For a reasoning sequence $\mathbf{x} = (x_1, x_2, \ldots, x_T)$,

$$\pi(\mathbf{x} \mid \mathbf{c}) = \prod_{t=1}^{T} \pi(x_t \mid \mathbf{x}_{<t}, \mathbf{c}), \qquad \pi(x_t \mid \mathbf{x}_{<t}, \mathbf{c}) = \pi_{\text{LM-head}}(x_t \mid z_t), \tag{1}$$

where $\mathbf{x}_{<t}$ denotes the sequence of tokens preceding position $t$ and $z_t := \pi_{\text{Transformer}}(\mathbf{x}_{<t}, \mathbf{c})$ is the latent representation associated with $x_t$. Following Hao et al. (2024), we refer to $z_t$ as lying in the latent space of $x_t$. At test time, the ground truth is unknown, and thus a reward function $R(\mathbf{x}, \mathbf{c})$ is introduced to evaluate the reasoning token sequence. Concluding the above, the objective for a reasoning problem is to find an optimal reasoning path:

$$\mathbf{x}^* = \arg\max_{\mathbf{x}} R(\mathbf{x}, \mathbf{c}). \tag{2}$$

Please refer to Section B for examples.

### 2.2 REASONING VIA POLICY GRADIENT IN LATENT SPACE

To solve the problem in Equation (2), we reformulate the task as optimizing over a sequence of latent representations rather than directly searching for tokens. Specifically, we denote a sequence of latent representations $\mathbf{z} = (z_1, z_2, \ldots, z_N)$, where $z_t$ lies in the latent space of $x_t$ and $N$ is typically an integer smaller than or equal to $T$. To identify the optimal sequence of latent representations, the to-be-optimized objective is as follows:

$$\mathbf{z}^* = \arg\max_{\mathbf{z}} \mathbb{E}_{\mathbf{x} \sim \pi(\mathbf{x}|\mathbf{z},\mathbf{c})}[R(\mathbf{x}, \mathbf{c})]. \tag{3}$$

To sample $\mathbf{x} \sim \pi(\mathbf{x} \mid \mathbf{z}, \mathbf{c})$, we first decode the latents $\mathbf{z}$ into their corresponding tokens, then continue with autoregressive generation, since the complete sequence may extend beyond the latent-initialized reasoning path. As $T$ is the complete sequence length, the factorization is

$$\pi(\mathbf{x} \mid \mathbf{z}, \mathbf{c}) = \underbrace{\prod_{t=1}^{N} \pi_{\text{LM-head}}(x_t \mid z_t)}_{\text{decode from latents}} \underbrace{\prod_{t=N+1}^{T} \pi(x_t \mid \mathbf{x}_{<t}, \mathbf{c})}_{\text{continue generation}}, \tag{4}$$

where, if the generation stops at $t = N$, the second product is empty and equals 1.

**Test-Time Optimization of Latent Representations.** Given the objective in Equation (3), we optimize the latent representations using a direct policy gradient approach based on REINFORCE (Williams, 1992). Assuming the *independence of the latent representations*, the update process is:

$$\mathbf{z} \leftarrow \mathbf{z} + \eta \nabla_{\mathbf{z}} \mathcal{J}(\mathbf{z}), \tag{5}$$

where the gradient $\nabla_{\mathbf{z}} \mathcal{J}(\mathbf{z})$, of our objective with respect to $\mathbf{z}$ can be derived as:

$$\nabla_{\mathbf{z}} \mathcal{J}(\mathbf{z}) = \mathbb{E}_{\mathbf{x} \sim \pi(\mathbf{x}|\mathbf{z},\mathbf{c})} \left[ R(\mathbf{x}, \mathbf{c}) \nabla_{\mathbf{z}} \log \pi(\mathbf{x} \mid \mathbf{z}, \mathbf{c}) \right]. \tag{6}$$

Leveraging Equation (4), the gradient of the $t$-th latent representation is:

$$[\nabla_{\mathbf{z}} \mathcal{J}(\mathbf{z})]_t = \mathbb{E}_{\mathbf{x} \sim \pi(\mathbf{x}|\mathbf{z},\mathbf{c})} \left[ R(\mathbf{x}, \mathbf{c}) \nabla_{z_t} \log \pi(x_t \mid z_t) \right], \tag{7}$$

where the expectation is approximated using the empirical mean in practical implementation. The independence assumption is adopted for two main reasons. (1) **Enlarged Exploration Space.** Without this assumption, the autoregressive structure would force the optimization to concentrate on the *first*

latent token, since all subsequent tokens are conditioned on it. This collapses the effective search space. By contrast, imposing independence decouples the latent components, yielding a substantially larger exploration space and a more flexible "launch pad" for generating the subsequent reasoning trajectory $\prod_{t=N+1}^{T} \pi(x_t \mid x_{<t}, \mathbf{c})$.(2) **Theoretical Guidance.** Our analysis in Section C draws an analogy to Multi-Prover Interactive Proofs (MIP), and shows that the potential loss introduced by the independence assumption can be mitigated by both the base model's ability to faithfully generate the follow-up reasoning trajectory and the reward model's ability to evaluate it. Please refer to Sections C and D for a theoretical ~~justification~~ guidance of this assumption and a detailed derivation of the policy-gradient update.

## 2.3 LATENTSEEK ALGORITHM

---

**Algorithm 1** LATENTSEEK

---

**Require:** Problem $\mathbf{c}$, learning rate $\eta$, pre-trained model $\pi$, reward threshold $\tau$, fraction ratio $\rho$, max iterations $K$

$\mathbf{x}, \mathbf{z} \leftarrow \pi(\mathbf{x} \mid \mathbf{c})$                     ▷ Initialize latent representations with CoT: Equation (1).

$r \leftarrow R(\mathbf{x}, \mathbf{c})$                                   ▷ Self Reward with Equation (8).

$T \leftarrow |\mathbf{x}|$

$\mathbf{z} \leftarrow [z_1, z_2, \ldots, z_{\rho T}]$                           ▷ Keep a $\rho$ fraction, such as 20%

**while** $k \leq K$ and $r \leq \tau$ **do**                      ▷ $K$ is typically 10

    $\mathbf{z} \leftarrow \mathbf{z} + \eta \nabla_{\mathbf{z}} \mathcal{J}(\mathbf{z})$           ▷ Update $\nabla_{\mathbf{z}} \mathcal{J}(\mathbf{z})$: Equations (6) and (7).

    $\mathbf{x} \sim \pi(\mathbf{x}|\mathbf{z}, \mathbf{c})$                    ▷ Sample sequence: Equation (4).

    $r \leftarrow R(\mathbf{x}, \mathbf{c})$                       ▷ Self Reward with Equation (8).

**end while**

**return** $\tilde{\mathbf{x}}$

---

The LATENTSEEK algorithm is summarized in Algorithm 1. At a high level, it performs an instance-level, gradient-guided search over a latent space by iteratively refining latent representations and querying a self-rewarding mechanism. Concretely, at iteration $k$, the current latents $\mathbf{z}$ are decoded to a full sequence $\mathbf{x}$ through a sampling method. Conditioned on $\mathbf{x}$ and the instance context $\mathbf{c}$, a scalar reward is then obtained from a self-rewarding mechanism:

$$R(\mathbf{x}, \mathbf{c}) \sim \pi(\cdot \mid \mathbf{x}, \mathbf{c}, \text{prompt}_{\text{self-reward}}). \tag{8}$$

This reward signal drives a gradient-guided refinement step that updates the latent representations, thereby biasing subsequent decodes toward higher-reward reasoning paths. The process runs for a small number of iterations (typically less than 3), stopping early if the reward exceeds a threshold.

**Enhancing Techniques.** To ensure more stable and efficient optimization, we propose the integration of two complementary techniques. First, *CoT Initialization*: The initial latent representation is derived from the CoT reasoning sequence. This approach leverages the reasoning capabilities of CoT to establish an effective starting point for subsequent optimization. Second, *Fractional Sequence Optimization*: Instead of optimizing the entire sequence of latent representations $\mathbf{z} = [z_1, z_2, \ldots, z_T]$, we propose to optimize only a subsequence $[z_1, z_2, \ldots, z_{\rho T}]$, where $\rho \in (0, 1]$ is a hyperparameter. The hyperparameter $\rho$ must strike a balance between two competing objectives: maintaining adequate representational capacity to support effective exploration, and limiting the number of latent representations being updated. Excessive modification of latent representations—especially when their decoded outputs lack semantic coherence—can compromise the reliability of the reward function.

## 3 EMPIRICAL RESULTS

### 3.1 EXPERIMENTAL SETUP

In this subsection, we list our experimental setup. See Sections F.1 to F.3 and F.5 to F.7 for more details.

**Reward Mechanism** We employ the mathematical reasoning prompts introduced by Lifshitz et al. (2025) to elicit the model's self-reward computation. The reward prompts remain consistent across all

Table 1: Accuracy results. **Bold**: best performance, Underlined: second-best. Numbers highlighted in red indicate the performance gap relative to the CoT method. Each rollout baseline uses a rollout number comparable to LATENTSEEK 's update iterations for fairness. See Section E.1 for larger-number comparisons, Section E.3 for token consumption, Section E.8 for performance trends across different fraction ratios, and Figure 3 for results of Qwen3 on AIME2024. $^{\dagger}$: requires training.

| Model / Methods | Qwen2 7B | Qwen2.5 1.5B | 7B | 14B | LLaMA3.1 8B | Avg |
|---|---|---|---|---|---|---|
| *GSM8K: Prompt 1* | | | | | | |
| CoT | 68.01 | 68.08 | 88.86 | 92.03 | 50.19 | 73.43 |
| BoN | 79.76 | 68.31 | 89.08 | 92.27 | 72.93 | 80.47 |
| Self-Reflection (Shinn et al., 2023) | 67.85 | 68.84 | 88.48 | 92.21 | 52.16 | 73.91 |
| Self-Consistency | 73.24 | 69.07 | 89.23 | 92.19 | 55.34 | 75.81 |
| SFT$^{\dagger}$ | 65.86 | 49.20 | 72.55 | 82.39 | 40.33 | 62.07 |
| SFT (Magpie 25K)$^{\dagger}$ | 76.50 | 66.48 | 83.01 | 90.30 | 70.81 | 77.42 |
| iCoT$^{\dagger}$ (Deng et al., 2024) | 47.54 | 23.28 | 41.02 | - | 47.08 | 39.73 |
| **LATENTSEEK** (*Self*) | **84.38**$_{+16.37}$ | **70.89**$_{+2.81}$ | **90.14**$_{+1.28}$ | **92.49**$_{+0.46}$ | **78.54**$_{+28.35}$ | **83.29**$_{+9.86}$ |
| *GSM8K: Prompt 2* | | | | | | |
| CoT | 65.20 | 15.31 | 66.41 | 91.81 | 69.07 | 61.56 |
| BoN | 61.33 | 6.14 | 74.04 | 92.27 | 75.97 | 61.95 |
| Self-Reflection (Shinn et al., 2023) | 68.16 | 27.98 | 72.63 | 91.96 | 76.22 | 67.39 |
| Self-Consistency | 64.37 | 8.79 | 69.22 | 92.49 | 74.30 | 61.83 |
| **LATENTSEEK** (*Self*) | **80.21**$_{+15.01}$ | **44.20**$_{+28.89}$ | **85.06**$_{+18.65}$ | **92.72**$_{+0.91}$ | **83.70**$_{+14.63}$ | **77.18**$_{+15.62}$ |
| *MATH-500: Prompt 1* | | | | | | |
| CoT | 51.40 | 54.80 | 72.80 | 77.20 | 47.60 | 60.76 |
| BoN | 53.40 | 47.40 | 75.40 | 78.80 | 51.20 | 61.24 |
| Self-Reflection (Shinn et al., 2023) | 53.00 | 53.00 | 72.40 | 75.80 | 45.40 | 59.92 |
| Self-Consistency | 53.60 | 51.60 | 73.20 | 77.40 | 50.40 | 61.24 |
| SFT (Magpie 25K)$^{\dagger}$ | 46.60 | 44.40 | 55.40 | 68.20 | 31.00 | 49.12 |
| **LATENTSEEK** (*Self*) | **57.40**$_{+6.00}$ | **55.60**$_{+0.80}$ | **75.60**$_{+2.80}$ | **80.00**$_{+2.80}$ | **54.60**$_{+7.00}$ | **64.64**$_{+3.88}$ |
| *MATH-500: Prompt 2* | | | | | | |
| CoT | 37.40 | 29.40 | 53.80 | 68.00 | 40.40 | 45.80 |
| BoN | 41.60 | 29.40 | 55.80 | 64.20 | 44.40 | 47.08 |
| Self-Reflection (Shinn et al., 2023) | 43.00 | 32.20 | 55.60 | 70.00 | 35.20 | 47.20 |
| Self-Consistency | 43.20 | 17.20 | 57.00 | 65.80 | 45.20 | 45.68 |
| **LATENTSEEK** (*Self*) | **44.80**$_{+7.40}$ | **34.40**$_{+5.00}$ | **57.60**$_{+3.80}$ | **71.00**$_{+3.00}$ | **47.00**$_{+6.60}$ | **50.96**$_{+5.16}$ |
| *AIME2024: Prompt1* | | | | | | |
| CoT | 0.00 | 3.33 | 6.67 | 10.00 | 0.00 | 4.00 |
| BoN | 0.00 | 0.00 | 10.00 | **16.67** | 0.00 | 5.33 |
| Self-Reflection (Shinn et al., 2023) | 0.00 | **6.67** | 6.67 | 10.00 | 0.00 | 4.67 |
| Self-Consistency | **3.33** | 0.00 | 3.33 | 13.33 | 6.67 | 5.33 |
| SFT (Magpie 25K)$^{\dagger}$ | **3.33** | 0.00 | 3.33 | 10.00 | 3.33 | 4.00 |
| **LATENTSEEK** (*Self*) | **3.33**$_{+3.33}$ | **6.67**$_{+3.33}$ | **13.33**$_{+6.67}$ | **16.67**$_{+6.67}$ | **10.00**$_{+10.00}$ | **10.00**$_{+6.00}$ |
| *AIME2024: Prompt2* | | | | | | |
| CoT | 0.00 | 0.00 | 0.00 | 3.33 | 0.00 | 0.67 |
| BoN | **3.33** | 0.00 | 6.67 | **10.00** | **6.67** | 5.33 |
| Self-Reflection (Shinn et al., 2023) | 0.00 | **3.33** | 0.00 | 6.67 | **6.67** | 3.33 |
| Self-Consistency | **3.33** | 0.00 | 3.33 | 3.33 | 0.00 | 2.00 |
| **LATENTSEEK** (*Self*) | **3.33**$_{+3.33}$ | **3.33**$_{+3.33}$ | **13.33**$_{+13.33}$ | **10.00**$_{+6.67}$ | **6.67**$_{+6.67}$ | **7.33**$_{+6.67}$ |

datasets. For some models on specific tasks, we further incorporate a format-based reward (DeepSeek-AI, 2025) to enhance their instruction-following capability. To further analysis the potential of our paradigm, we introduce a Perfect Sharp Reward Model (PSRM), the details of which are discussed in §3.3. All generated reward is chosen to be a number between -1 and 0.

**Prompt Designation** For robustness consideration, we use two prompts for evaluation, with the first requiring a wrapped answer with $\backslash boxed\{\}$ (Yang et al., 2024; Team, 2024a) (Prompt 1) and the second asking to format the answer as a json (Prompt 2).

**Backbones** To evaluate the generalizability of our approach, we conduct experiments with pre-trained LLMs of different families and sizes: Qwen2-7B-Instruct, Qwen2.5-1.5B, 7B, 14B-Instruct (Yang et al., 2024; Team, 2024a), LLaMA3.1-8B-Instruct (Team, 2024b).

**Benchmarks** Following Deng et al. (2024); Liu et al. (2025), we focus on mathematical reasoning for evaluation. We evaluate on two standard benchmarks for reasoning, GSM8K (Cobbe et al., 2021) and MATH-500 (Hendrycks et al., 2021), and a harder dataset AIME2024.

Table 2: Accuracy score compared with more baseline methods on GSM8K and MATH-500 datasets with Llama3.1-8B as backbone. The best performances are highlighted in bold.

| Methods | Supervision | Train | Backbone | GSM8K | MATH-500 | Average |
|---|---|---|---|---|---|---|
| CoT | - | ✗ | Instruct | 69.1 | 47.6 | 58.3 |
| Few-Shot CoT (Lambert et al., 2025) | - | ✗ | Instruct | 83.4 | 42.5 | 63.0 |
| BoN | Self | ✗ | Instruct | 76.0 | 51.2 | 63.6 |
| Self-Rewarding (Yuan et al., 2025) | Self | ✔ | Instruct | 76.0 | 30.2 | 53.1 |
| Self-Relfection (Shinn et al., 2023) | Self | ✗ | Instruct | 76.2 | 45.4 | 60.8 |
| Self-Consistency | Self | ✗ | Instruct | 55.3 | 48.2 | 51.8 |
| ScPO (Prasad et al., 2024) | Self | ✔ | Instruct | 71.1 | 31.0 | 51.1 |
| CoH (Liu et al., 2023a) | Self | ✔ | Instruct | 74.4 | 32.3 | 53.3 |
| Genius (Xu et al., 2025) | Self | ✔ | Instruct | 78.3 | 34.6 | 56.5 |
| SPIN (Chen et al., 2024a) | Data | ✔ | Instruct | 74.9 | 31.5 | 53.2 |
| iCoT (Deng et al., 2024) | Data | ✔ | Instruct | 47.1 | - | 47.1 |
| SFT (Magpie 25K) | Data | ✔ | Instruct | 70.8 | 31.0 | 50.9 |
| GRPO (GSM8K Train) | Data | ✔ | Instruct | - | 50.2 | 50.2 |
| SimpleRL-Zoo (Zeng et al., 2025) | Data | ✔ | Base | 79.2 | 23.0 | 51.1 |
| **LATENTSEEK** | Self | ✗ | Instruct | **83.7** | **54.6** | **69.2** |

**Baselines**   We compare our methods against several established baselines:

- **Prompting (Training-Free)**: CoT (Wei et al., 2022) and Few-Shot CoT (Lambert et al., 2025).
- **Explicit Search (Training-Free)**: Best-of-N (BoN) represents a highly effective search strategy, as demonstrated by Liu et al. (2025). Self-Reflection (Shinn et al., 2023) is a search strategy imitating humans' reasoning pattern. Self-Consistency, also named majority vote, is a search mechanism in explicit space that stems from a multi-agent system.
- **Reinforcement Learning**: (1) Self Reward: Self-Rewarding (Yuan et al., 2025), ScPO (Prasad et al., 2024), CoH (Liu et al., 2023a), and Genius (Xu et al., 2025). (2) Verifiable Reward: SimpleRL-Zoo (Zeng et al., 2025), GRPO (GSM8K train set), and SPIN (Chen et al., 2024a).
- **Latent Chain-of-Thought**: Although iCoT (Deng et al., 2024) requires an augmented training dataset for GSM8K (Cobbe et al., 2021), it remains a canonical example of the latent CoT.
- **Supervised Fine-Tuning (SFT)**: Following (Xu et al., 2025), we apply SFT on Magpie 25K. For GSM8K, we also report SFT performance using its provided training set.

**Sampling Method**   We use greedy decoding to sample from LATENTSEEK, as it is more computationally efficient and more robust. (See Section E.4 for an experimental comparison).

## 3.2   STATE-OF-THE-ART TEST-TIME REASONING PERFORMANCE

The main experimental results are presented in Table 1 and Table 2.

**Best Performance on GSM8K (Cobbe et al., 2021) and MATH-500 (Hendrycks et al., 2021).** As shown in Table 1, LATENTSEEK surpasses every baseline across all 30 settings and achieves the best average performance over all datasets. When averaged across backbones, the improvement over *BoN* is **15.23** points on GSM8K and against *Self-Reflection* is **4.72** points on MATH-500. Compared with *Self-Consistency*, the gap is **15.35** on GSM8K and **5.28** on MATH-500. The Qwen2.5 series was explicitly trained with Prompt-1–distilled data (Team, 2024a), yielding strong *CoT* performance; nevertheless, LATENTSEEK achieves further improvements and attains the highest scores across baselines. Using LLaMA3.1 as the backbone, as shown in Table 2, we compare against a broader set of baselines. On GSM8K, our method surpasses the *CoT* baseline by **14.6** points and *BoN* by **7.7** points; on MATH-500, the improvements are **7.0** and **3.4** points, respectively. It also outperforms the self-reward RL method *Genius* (Xu et al., 2025) by **5.4** points on GSM8K and **20.0** points on MATH-500. Even compared with verifiable-feedback RL methods such as *SimpleRL-Zoo* (Zeng et al., 2025), trained on the base model, LATENTSEEK leads by an average of **18.1** points across the two datasets.

**Robustness Across Backbones.**   The robustness of our method across backbones can be evaluated along two axes: (1) *Model Families*: LATENTSEEK delivers best results across different models. Focusing on 7–8B models and using the same averaging protocol over GSM8K, MATH-500, and

Table 3: Accuracy results with Perfect Sharp Reward Model (PSRM). *Self*: the self-reward mechanism. The reported score is the average of both prompts.

| Reward | Qwen2 7B | 1.5B | Qwen2.5 7B | 14B | LLaMA3.1 8B | Avg |
|---|---|---|---|---|---|---|
| *GSM8K* | | | | | | |
| LATENTSEEK-*Self* | 82.3 | 57.5 | 87.6 | 92.6 | 81.1 | 80.2 |
| LATENTSEEK-*Perfect Sharp Reward Model* | $92.8_{+10.5}$ | $76.5_{+19.0}$ | $93.9_{+6.3}$ | $96.0_{+3.4}$ | $91.7_{+10.6}$ | $90.2_{+10.0}$ |
| *MATH-500* | | | | | | |
| LATENTSEEK-*Self* | 51.1 | 45.0 | 66.6 | 75.5 | 50.8 | 57.8 |
| LATENTSEEK-*Perfect Sharp Reward Model* | $71.8_{+20.7}$ | $66.7_{+21.7}$ | $81.8_{+15.2}$ | $86.6_{+11.1}$ | $70.5_{+19.7}$ | $75.5_{+17.7}$ |
| *AIME2024* | | | | | | |
| LATENTSEEK-*Self* | 3.3 | 5.0 | 13.3 | 13.3 | 8.3 | 8.6 |
| LATENTSEEK-*Perfect Sharp Reward Model* | $8.3_{+5.0}$ | $6.7_{+1.7}$ | $15.0_{+1.7}$ | $25.0_{+11.7}$ | $10.0_{+1.7}$ | $13.0_{+4.4}$ |

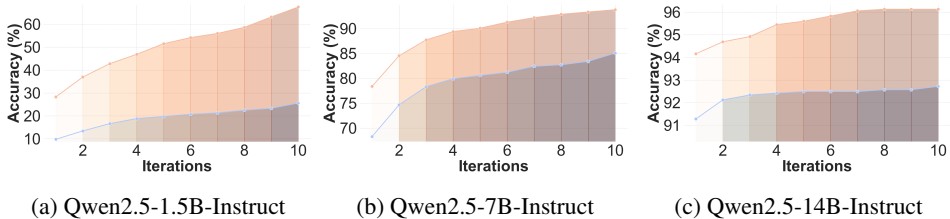

| (a) Qwen2.5-1.5B-Instruct | (b) Qwen2.5-7B-Instruct | (c) Qwen2.5-14B-Instruct |

Figure 2: GSM8K(Cobbe et al., 2021) Prompt 2 Accuracy changes with respect to the increasing number of iterations. Orange: Perfect Sharp Reward Model. Blue: Self Reward Model.

both prompts, LATENTSEEK surpasses *BoN* by **7.67** points on Qwen2, exceeds *Self-Reflection* by **4.82** on Qwen2.5, and improves over *Self-Consistency* by **9.65** on LLaMA3.1. (2) *Model Size*: At 1.5B parameters, LATENTSEEK surpasses *BoN* on GSM8K by **28.89** points and at 14B parameters, the margin on MATH-500 attains **6.80** points. These results provide robust evidence of our method's robustness across diverse model families and scale.

**Challenging Problems — AIME2024.** AIME2024 results appear in Table 1. Our method consistently outperforms all baselines, achieving an average gain of **6.34** points over *CoT* across model families and prompt configurations. The largest improvement occurs on AIME2024 with Prompt 2, where our approach exceeds *CoT* by **13.33** points and outperforms *BoN* by **6.67** points using Qwen2.5-7B-Instruct. Averaged over backbones, LATENTSEEK also surpasses *BoN* by **4.67** points, *Self-Reflection* by **5.33** points, and *Self-Consistency* by **4.67** points, indicating strong effectiveness in more complex settings. For the performance of a stronger model (Qwen3-4B-Instruct-2507 (Yang et al., 2025)) on AIME2024, please refer to Section E.2.

## 3.3 IDEAL EXPERIMENT: PERFECT SHARP REWARD MODEL

To inspect the value of exploration in latent space, we evaluate a Perfect Sharp Reward Model (PSRM) that returns 0 only when the final answer exactly matches the ground truth and $-1$ otherwise. This yields an all-or-nothing signal: until the correct answer is reached, every step receives identical feedback. Results (Table 3) show that, despite this extreme sharpness, the PSRM still delivers substantial improvements, outperforming the self-reward variant by an average of 10.67 points across all evaluated settings. Conceptually, this setup reduces optimization to maximizing the success rate under a near-constant reward landscape; the search carries almost no directional information and is therefore close to blind exploration. This demonstrate that **exploration in latent space alone can yield meaningful performance improvements, even under sharp feedback**.

**Small Model and Large Iterations.** Following Liu et al. (2025), we analyze the small Qwen2.5-1.5B-Instruct (Team, 2024a) under a large iteration budget to probe the limits of latent-space exploration. As shown in Table 4, Qwen2.5-1.5B-Instruct yields a **14-point** gain over GPT-4o on AIME2024. On MATH-500, it achieves a high accuracy score, trailing o1-preview by only **2.7** points. These results demonstrate that **even a 1.5B-parameter model's latent space is highly expressive and that LATENTSEEK can be an effective mechanism for eliciting this expressivity**.

Table 4: Performance of Extreme Scaling on MATH-500 (Hendrycks et al., 2021) and AIME2024. Setting the maximum update iteration to 256. $K$: average number of outputs or iterations.

| Model | Size | Type | MATH-500 | | AIME2024 | |
|---|---|---|---|---|---|---|
| | | | Acc | K | Acc | K |
| GPT-4o | - | CoT | 74.6 | - | 9.3 | - |
| o1-preview | - | CoT | 85.5 | - | 44.6 | - |
| Qwen2.5-1.5B-Inst. | 1.5B | TTS (Liu et al., 2025) | 81.8 | 256.0 | 20.0 | 256.0 |
| LATENTSEEK *(PSRM)* (Qwen2.5-1.5B-Inst.) | 1.5B | Latent Search | 82.8 | 61.8 | 23.3 | 211.8 |

### 3.4 TEST-TIME SCALING: SCALING TREND ANALYSIS OF LATENTSEEK

In addition to increasing the number of generated tokens at test time (Muennighoff et al., 2025; Snell et al., 2025), we propose an alternative approach to test-time scaling: increasing the number of LATENTSEEK iterations. The relationship between model performance on GSM8K (Cobbe et al., 2021) and the number of iterations is illustrated in Figure 2. As shown, an ideal reward model demonstrates strong performance and yields a consistently improving trend with more iterations. In contrast, the self-reward method exhibits rapid initial gains followed by a plateau, a behavior we attribute to the reward model's limited accuracy. Notably, as the number of iterations increases further, performance eventually surpasses this plateau and continues to improve. The above shows that **test-time scaling remains attainable even under a sharply peaked reward**. It further suggests that, given an appropriate reward model, **searching through the latent space offers a promising new direction for test-time scaling.**

### 3.5 COMPARISON WITH UNGUIDED SEARCH

We compare LATENTSEEK with a single-step stochastic explorative search in latent space on the GSM8K (Cobbe et al., 2021), to assess the effectiveness of our gradient-guided search. In this baseline, instead of performing iterative gradient ascent, a Gaussian noise is added once to the initial latent representations,

$$z \leftarrow z + \epsilon, \ \epsilon \sim \mathcal{N}(0, \sigma^2 I)$$

, and evaluate three noise levels $\sigma^2 = 0.5, 0.75,$ and $1.0$. As shown in Table 5, LATENTSEEK outperforms the

Table 5: Comparison with Unguided Search on GSM8K. *SE*: Stochastic Exploration.

| Model | Qwen2 | Qwen2.5 | | | LLaMA3.1 | Avg |
|---|---|---|---|---|---|---|
| | 7B | 1.5B | 7B | 14B | 8B | |
| *GSM8K: Prompt 1* | | | | | | |
| SE ($\sigma^2 = 0.50$) | 66.19 | 64.22 | 88.60 | 92.33 | 49.73 | 72.21 |
| SE ($\sigma^2 = 0.75$) | 65.96 | 63.53 | 88.60 | 92.03 | 49.73 | 71.97 |
| SE ($\sigma^2 = 1.00$) | 65.13 | 63.76 | 89.08 | 91.88 | 50.57 | 72.08 |
| LATENTSEEK | 84.38 | 70.89 | 90.14 | 92.49 | 78.54 | 83.29 |
| *GSM8K: Prompt 2* | | | | | | |
| SE ($\sigma^2 = 0.50$) | 64.32 | 16.68 | 65.60 | 92.04 | 67.85 | 61.30 |
| SE ($\sigma^2 = 0.75$) | 64.77 | 15.69 | 64.92 | 91.89 | 67.02 | 60.86 |
| SE ($\sigma^2 = 1.00$) | 63.84 | 16.07 | 66.34 | 92.12 | 66.72 | 61.02 |
| LATENTSEEK | 80.21 | 44.20 | 85.06 | 92.72 | 83.70 | 77.18 |

stochastic exploration baselines by an average of **13.66 points**, indicating that **the gains arise from effective gradient guidance rather than random perturbations.**

### 3.6 ANALYSIS OF SELF-REWARD MECHANISM

To understand the role of the self-reward mechanism, we evaluate the accuracy of the self-reward mechanism on the GSM8K test set, in which the generation uses the prompt 2 (the json prompt). Specifically, we collect *all explored sequences*, including intermediate reasoning steps as well as

Table 6: Self-Reward Mechanism Percentage Accuracy on GSM8K.

| Backbone | Qwen2 | Qwen2.5 | | | LLaMA3.1 | Avg |
|---|---|---|---|---|---|---|
| | 7B | 1.5B | 7B | 14B | 8B | |
| Acc | 85.78 | 77.15 | 80.17 | 89.83 | 80.46 | 82.68 |

final generated answers, and evaluate the alignment between the reward and ground-truth correctness. We binarized our dense $[-1, 0]$ reward to "Positive" (Reward $> -0.2$) and "Negative" (Reward

$\le -0.2$) and compared this against the ground-truth correctness. This yields the results in Table 6. The table demonstrates that in approximately 80% of the cases, the self-reward mechanism is reliable. In practice, the ability to judge an answer cooperates with the ability to generate one, and together they lead to substantially improved overall performance.

Furthermore, we replace our self-reward mechanism with a constant negative reward baseline to isolate the effect of reward guidance, assigning **a constant reward of -1 at every step.** The experiment is performed on the same condition as Table 6. A negative reward is used because any non-negative reward would simply reinforce the current tokens under argmax decoding and hinder exploration. As reported in Table 7, this ablation causes **an average performance drop of 14.21 points**(eg., 85.06% to 64.52% for Qwen2.5-7B-Instruct), demonstrating that **our self-reward guidance is essential for effective optimization.**

Table 7: Comparison with `Constant Reward` baseline

| Backbone | Qwen2 7B | Qwen2.5 1.5B | Qwen2.5 7B | Qwen2.5 14B | LLaMA3.1 8B | Avg |
|---|---|---|---|---|---|---|
| LATENTSEEK (Constant) | 60.19 | 29.49 | 64.52 | 90.67 | 69.98 | 62.97 |
| LATENTSEEK (Self) | **80.21** | **44.20** | **85.06** | **92.72** | **83.70** | **77.18** |
| Δ | +20.02 | +14.71 | +20.54 | +2.05 | +13.72 | +14.21 |

### 3.7 QUALITATIVE ANALYSIS

An analysis of the LATENTSEEK's generated responses offers insight into its reasoning processes, with a representative example detailed in Table 8. These responses often contain nonsensical or obscure tokens—such as "andLet", "on80ward)", and "BothThe"—as well as syntactically absurd phrases like "Let find this" and "denote the common". This phenomenon indicates that the reasoning processes of pre-trained language models can diverge significantly from human cognition, even when they produce a correct final output. Consequently, this suggests that **optimal reasoning paths for language models may not align with human strategies, and thus may be more effectively sought within the latent space**, which is native to LLM and therefore also a natural choice. A more detailed qualitative analysis and additional cases are presented in Section I.

Table 8: Case Study. Latent optimized tokens are in blue. Please refer to Section I for more cases.

> QUESTION: Two trains leave San Rafael at the same time [...] What's the distance covered by each train in the two days?
> ANSWER: 230
> LATENTSEEK: Let find this,, let'll more understand it down step two andLet 1: BothThe the trains are same route west both first time on80ward) on same and and can consider they the travel travel same distance of So's denote the common xd' they both 80 miles on x have write up an following: [...] [...] Since both trains travel the same distance in each direction, the distance covered by each train is 230 miles. The final answer is: \boxed{230}

### 3.8 ALGORITHMIC STATISTICS

Table 9 reports two key statistical features: (1) the ratio of final answer tokens to the original CoT tokens, and (2) the average number of update iterations across the entire dataset. The ratio of the final answer's token count to that of the original CoT rarely exceeds 1.1, confirming that performance gains are not achieved through verbose generation. Moreover, the method converges rapidly, requiring an average of fewer than two update iterations for average-level questions. This highlights the computational efficiency and rapid convergence behavior of our method.

Table 9: Algorithmic Statistics: (1) The ratio of answer length to CoT length. (2) Average update iterations.

| Model | Qwen2 7B | Qwen2.5 1.5B | Qwen2.5 7B | Qwen2.5 14B | LLaMA3.1 8B | Avg |
|---|---|---|---|---|---|---|
| *GSM8K: Prompt 1* | | | | | | |
| \|Answer\|/\|CoT\| | 0.97 | 0.99 | 1.00 | 1.00 | 0.94 | 0.98 |
| Avg.# Iter | 0.94 | 0.75 | 0.24 | 0.05 | 1.66 | 0.72 |
| *GSM8K: Prompt 2* | | | | | | |
| \|Answer\|/\|CoT\| | 1.06 | 3.80 | 1.08 | 1.00 | 1.02 | 1.59 |
| Avg.# Iter | 0.55 | 4.59 | 1.52 | 0.14 | 0.58 | 1.48 |
| *MATH-500: Prompt 1* | | | | | | |
| \|Answer\|/\|CoT\| | 0.92 | 0.97 | 0.99 | 0.99 | 0.92 | 0.96 |
| Avg.# Iter | 2.09 | 1.93 | 1.10 | 0.62 | 2.35 | 1.62 |
| *MATH-500: Prompt 2* | | | | | | |
| \|Answer\|/\|CoT\| | 1.01 | 0.94 | 0.98 | 0.97 | 0.90 | 0.96 |
| Avg.# Iter | 0.72 | 2.19 | 0.98 | 0.49 | 1.65 | 1.21 |

To further demonstrate the value of shifting computation from training to test time, we compare Floating Point Operations (FLOPs) against Genius (Xu et al., 2025), a strong RL-based self-rewarding

baseline. As shown in Table 2, LATENTSEEK outperforms Genius by 12.7% on GSM8K using LLaMA3.1-8B as the backbone. For this model, a single forward pass costs approximately $2.29 \times 10^{11}$ FLOPs. Crucially, Genius requires full backpropagation through the entire model to update parameters, whereas LATENTSEEK backpropagates only through the language model, incurring about $1.05 \times 10^{9}$ FLOPs. Following our calculations, LATENTSEEK remains more efficient for processing up to approximately $1.94 \times 10^{5}$ inference instances when compared to the total cost of the Genius framework. This threshold notably exceeds the 100k examples in Genius's training set, underscoring the significant pratical efficiency of our approach. Please refer to Section G for detailed calculation.

## 4 RELATED WORK

**Reasoning in Language Models.**    Recent advances in reasoning capabilities of large language models (Brown et al., 2020; OpenAI, 2023) have been enhanced through prompting techniques. CoT prompting (Wei et al., 2022; Kojima et al., 2022; Zhou et al., 2022) encourages models to generate intermediate reasoning steps. Unlike these static approaches, our method dynamically optimizes the reasoning process for each problem instance. Compute-optimal scaling (Snell et al., 2025; Misaki et al., 2025) adaptively adjusts inference strategies based on task complexity. Latent reasoning methods (Hao et al., 2024; Shen et al., 2025; Cheng and Van Durme, 2024; Deng et al., 2024; Chen et al., 2024b) replace explicit text-based reasoning with continuous representations. The broader field of learning to reason includes techniques like process supervision (Uesato et al., 2022) and self-critique (Huang et al., 2022).

**Reinforcement Learning for Language Models.**    The integration of Reinforcement Learning and LLM starts from the realm of Human Feedback (RLHF) (Ouyang et al., 2022; Bai et al., 2022; Rafailov et al., 2023), which commonly employs algorithms such as Proximal Policy Optimization (PPO) (Schulman et al., 2017). Recent advancements (Liu et al., 2023b; Guo et al., 2025) specifically address reasoning tasks with RL techniques. Innovations in reward modeling have explored human preference-based architectures (Schulman et al., 2017; Hazra et al., 2024), automated language model-driven design (Kwon et al., 2023), and multi-agent verification frameworks (Lifshitz et al., 2025).

**Controllable Generation and Test-Time Optimization.**    Various approaches have been developed for controlling language model outputs, including conditioning on control codes (Keskar et al., 2019), gradient-based steering (Dathathri et al., 2019), and prompt optimization (Qin et al., 2023). At test time, techniques (Sun et al., 2023; Zhang et al., 2025; Wang et al., 2022) improve outputs through sampling and selection. Some Test-Time Training(TTT) framewroks (Sun et al., 2020; 2024; Hardt and Sun, 2023) integrate self-supervised objectives for online model updates. Our work differs from test-time planning approaches (Hao et al., 2023) by optimizing in a continuous latent space rather than performing a discrete search.

**Prompt Tuning and Soft Prompt.**    Prompt Tuning adapts language models by prepending trainable vectors to inputs or hidden states (Lester et al., 2021; Liu et al., 2024a; Li and Liang, 2021; Liu et al., 2021). However, both require labeled data and full backpropagation, incurring high computational cost. In contrast, our method leverages latent-space manipulation without training data or model updates, enabling efficient, flexible adaptation.

## 5 CONCLUSION

In conclusion, the LATENTSEEK framework introduces a novel and efficient approach to enhancing reasoning capabilities in LLMs by leveraging TTIA in the latent space. By optimizing latent representations through policy gradient, LATENTSEEK circumvents the need for parameter updates, offering an alternative to methods that require substantial retraining or reinforcement learning. Empirical results across multiple reasoning benchmarks consistently demonstrate the superior performance of LATENTSEEK compared to existing baselines, such as CoT, BoN, and reinforcement learning-based techniques. Furthermore, the framework proves to be computationally efficient, with rapid convergence for average-level problems. This work also demonstrates a new possible avenue for test-time scaling in the latent space. Ultimately, LATENTSEEK represents a significant step forward in advancing LLMs in the realm of TTIA reasoning. Please refer to Section A for discussions and future works.

## ETHICS STATEMENT

This work focuses on improving the reasoning capabilities of large language models (LLMs). The research is methodological in nature and does not involve human subjects, sensitive data, or deployment in real-world applications. Our contributions are confined to improving core algorithmic aspects of LLM reasoning and do not introduce new data that could raise concerns regarding privacy, bias, or misuse. While we recognize that LLMs can have broader societal impacts, particularly when used in downstream applications, our work does not directly engage with these deployment scenarios. We also note that enhanced reasoning capabilities may indirectly influence downstream model behavior. However, the improvements described in this paper are academic-purpose and do not facilitate manipulation, deception, or unethical use of LLMs. Overall, we believe that our research poses no direct ethical or societal risks and is aligned with the responsible development of trustworthy AI systems.

## REPRODUCIBILITY STATEMENT

The findings presented in this paper are supported by a detailed disclosure of our methodology, designed to enable full reproducibility. The core methodology and algorithmic formulation of our approach are presented in §2. Our full experimental protocol, which covers the datasets, evaluation benchmarks, and baselines, is detailed in §3. All requisite implementation details for replication, including model backbones and hyperparameters, are documented in Section F. Taken together, the paper and its appendices provide a complete blueprint for reproducing our work. For the statement on our use of LLMs, please see Section K. The code is attached to the submission.

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

# Appendices

## Contents

## A  DISCUSSION AND FUTURE WORKS

While our work demonstrates the effectiveness of latent space optimization, we offer some discussion for inspiring future research.

**Reward Models.**  A primary limitation of our current approach is its reliance on a self-rewarding mechanism. While this proves effective, the optimization process is inherently constrained by the base model's own evaluation capabilities and potential biases, lacking a truly objective external signal. The ideal scenario would involve a principled Outcome Reward Model (ORM) that assesses the final answer and provides unambiguous guidance for the search. However, as detailed in our experiments in Section E.5, we found that current publicly available ORMs are not yet sufficiently robust or generalizable for this task; their noisy signals proved less effective for guiding the latent space optimization than the more consistent self-reward mechanism. **This highlights a critical area for future work: the development of more powerful verifiers capable of reliably scoring complex reasoning tasks.**

**Latent Optimization.**  We adopt standard policy-gradient methods in our implementation, leaving the exploration of more advanced reinforcement learning algorithms—such as Proximal Policy Optimization (PPO)—to future work. In addition, pursuing latent-space–specific optimization may be a worth exploring direction.

**Large Base Model.**  Our experiments are conducted on models up to 14B-parameter scale, constrained by available computational resources. Scaling the approach to larger base models remains an important avenue for future investigation.

## B  METHODS OF TEST-TIME INSTANCE-LEVEL REASONING

We list the formulations of two classical test-time instance-level reasoning methods:

- Prompt Engineering: Given the problem instance prompt $c$, the reward function is simply the language modeling distribution:

$$\mathbf{x}^* = \arg\max_{\mathbf{x}} \pi(\mathbf{x} \mid \mathbf{c}) \tag{9}$$

- Best-of-N (BoN): Given $N$ i.i.d sequences $\mathbf{x}_{(1)}, \mathbf{x}_{(2)}, \ldots, \mathbf{x}_{(N)} \sim \pi(\cdot \mid \mathbf{c})$,

$$\mathbf{x}^* = \arg\max_{\mathbf{x} \in \{\mathbf{x}_{(1)}, \mathbf{x}_{(2)}, \ldots, \mathbf{x}_{(N)}\}} R(\mathbf{x}, c) \tag{10}$$

## C  THEORETICAL ANALYSIS

This section offers a theoretical framework for LATENTSEEK, with a main focus on justifying the expressiveness of the independence among latent variables, which may be a good start for understanding and inspecting our algorithm.

### C.1  PRELIMINARIES: MULTIPLE PROVER INTERACTIVE PROOFS AND NEXP

This section introduces the concept of Multi-prover Interactive Proofs (MIP) and NEXP for completeness. A comprehensive exposition of MIP is provided in Vadhan and Zhang (2002), while Arora and Barak (2006) offers a detailed introduction to theoretical computer science, encompassing concepts such as language and Turing Machine.

In the Multi-Prover Interactive Proof (MIP) model, provers may communicate with one another prior to the initiation of the proof process. Once the proof process begins, however, such communication is prohibited, and each prover interacts with the verifier in a fully private manner.

**Definition C.1** (Multiple Prover Interaction). Let $P_1, P_2, \ldots, P_k : \{0,1\}^* \to \{0,1\}^*$ be functions. A $n$-round interactions between the verifier $V$ and the provers $P_1, P_2, \ldots, P_k$, denoted by $\langle (P_1, P_2, P_3, \ldots, P_k), V \rangle (x)$ is the sequence of the following strings $q_{11}, q_{21}, \ldots, q_{k1}, a_{11}, a_{21},$

$$\ldots, a_{k1}, q_{12}, \ldots, q_{k2}, a_{12}, \ldots, a_{k2}, \ldots, q_{1n}, q_{2n}, \ldots, q_{kn}, \ldots, a_{1n}, \ldots, a_{kn}, v \text{ defined as follows:}$$

$$q_{11}, q_{21}, \ldots, q_{k1} = V(x)$$
$$a_{11} = P_1(x, q_{11})$$
$$a_{21} = P_2(x, q_{21})$$
$$\ldots$$
$$a_{k1} = P_k(x, q_{k1})$$
$$q_{12}, q_{22}, \ldots, q_{k2} = V(x, q_{11}, a_{11}, \ldots, q_{k1}, a_{k1})$$
$$\ldots$$
$$a_{kn} = P_k(x, q_{k1}, a_{k1}, \ldots, q_{kn})$$
$$v = V(x, q_{11}, a_{11}, \ldots, a_{kn}, q_{kn})$$

We denote $\langle (P_1, P_2, P_3, \ldots, P_k), V \rangle_V (x)$ to be last output $v$.

**Definition C.2** ($k$-MIP Vadhan and Zhang (2002)). A language $L$ is in $k$-MIP if there is a Turing machine verifier $V$ such that on inputs $x, a_{11}, \ldots, a_{ij}$, $V$ runs in time polynomial in $|x|$ and such that:

- Efficiency: The number and length of all messages exchanged is at most polynomial in the common input $x$.

- Completeness: $x \in L \Rightarrow \exists P_1, P_2, \ldots, P_k, \ Pr[\langle (P_1, P_2, \ldots, P_k), V \rangle_V (x) = 1] \geq \frac{2}{3}$

- Soundness: $x \notin L \Rightarrow \forall P_1, P_2, P_3, \ldots, P_k, \ Pr(\langle (P_1, P_2, \ldots, P_k), V \rangle_V (x) = 1] \leq \frac{1}{3}$

**Definition C.3** (MIP Vadhan and Zhang (2002)). $\text{MIP} = \cup_k k - \text{MIP}$

Next, we'll introduce NEXP.

**Definition C.4** (NTIME Arora and Barak (2006)). For every function $T : \mathbb{N} \to \mathbb{N}$ and $L \subseteq \{0,1\}^*$, we say that $L \in \text{NTIME}(T(n))$ if there is a constant $c > 0$ and a $cT(n)$-time non-deterministic Turing Machine $M$ such that for every $x \in \{0,1\}^*$, $x \in L \Leftrightarrow M(x) = 1$.

**Definition C.5** (NP). $\text{NP} = \cup_{k \in \mathbb{N}} \text{NTIME}(n^k)$

**Definition C.6** (NEXP). $\text{NEXP} = \cup_{k \in \mathbb{N}} \text{NTIME}(2^{n^k})$

## C.2 THEORETICAL ANALYSIS: INDEPENDENT UPDATING

Mentioned in Section 2, the latent representations are independent, which results in the independence of the token update process, which at first glance may constrain the model's expressive capacity. In this subsection, we focus on decision problems Arora and Barak (2006). Despite discrepancies between theoretical frameworks and practical challenges, valuable insights can be gained. Specifically, it is possible to demonstrate that the model's expressivity remains theoretically comparable in spite of the updating independence.

Our method is strongly related to a TCS complexity class, Multi Prover Interactive Proofs Ben-Or et al. (1988) or simply MIP. We initially outline the notations for multiple prover interactions and subsequently define the associated complexity class for our latent thought framework, which we later demonstrate is equivalent to MIP and the complexity class NEXP (solved in exponential time by a non-deterministic Turing Machine). Please refer to Section C.1 for preliminaries. We list the relationship as follows:

- MIP: A verifier interacts with multiple non-communicating provers to decide membership in a language.
- LATENTSEEK: the optimized tokens, as they are updated independently, act like the non-communicated provers who jointly determine the outcome. After decoding the latent vectors, where each prover proposes its own output, the subsequent autoregressive generation is regarded as part of the verifier function $V$. Regarding the CoT initialization, recall that the MIP framework permits communication among provers *before* the proving process begins. Accordingly, initializing the latent vectors with a CoT prompt can be interpreted as this pre-proof communication stage.

- MIP-Bounded (LATENTSEEK): Different from MIP, each prover in LatentSeek can only generate 1 token, whose number of bits is bounded. Furthermore, a polynomial proportion of tokens (such as, $\rho = 0.2$) is allowed to be optimized, leading to a polynomial number of provers.

**Definition C.7** (Multiple Prover Interaction). Let $P_1, P_2, \ldots, P_k : \{0,1\}^* \to \{0,1\}^*$ be functions. A $n$-round interactions between the verifier $V$ and the provers $P_1, P_2, \ldots, P_k$, denoted by $\langle (P_1, P_2, P_3, \ldots, P_k), V \rangle (x)$ is the sequence of the following strings $q_{11}, q_{21}, \ldots, q_{k1}, a_{11}, a_{21}, \ldots, a_{k1}, q_{12}, \ldots, q_{k2}, a_{12}, \ldots, a_{k2}, \ldots, q_{1n}, q_{2n}, \ldots, q_{kn}, \ldots, a_{1n}, \ldots, a_{kn}, v$ defined as follows:

$$q_{11}, q_{21}, \ldots, q_{k1} = V(x)$$
$$a_{11} = P_1(x, q_{11})$$
$$a_{21} = P_2(x, q_{21})$$
$$\ldots$$
$$a_{k1} = P_k(x, q_{k1})$$
$$q_{12}, q_{22}, \ldots, q_{k2} = V(x, q_{11}, a_{11}, \ldots, q_{k1}, a_{k1})$$
$$\ldots$$
$$a_{kn} = P_k(x, q_{k1}, a_{k1}, \ldots, q_{kn})$$
$$v = V(x, q_{11}, a_{11}, \ldots, a_{kn}, q_{kn})$$

We denote $\langle (P_1, P_2, P_3, \ldots, P_k), V \rangle_V (x)$ to be last output $v$.

Different from MIP, which allows each prover to output a polynomial-length string, in our method, each prover can only output a bounded length of string (a token). We name the corresponding complexity class of our method as MIP-Bounded.

**Definition C.8** (MIP-Bounded). A language $L$ is in MIP-Bounded if there is a Turing machine verifier $V$, and a polynomial function poly$(\cdot)$ such that on inputs $x, a_{ij}, q_{ij}$, $V$ runs in time polynomial in $|x|$ and such that:

- Bounded: $\forall i$, output of $P_i$ is bounded, its output denoted as $|a_{ij}|$ satisfies that $|a_{ij}| \leq C$, where $C$ is a constant greater than 1.

- Completeness: $x \in L \Rightarrow \exists P_1, P_2, \ldots, P_{\text{poly}(|x|)}, Pr[\langle (P_1, P_2, \ldots, P_{\text{poly}(|x|)}), V \rangle_V (x) = 1] \geq \frac{2}{3}$

- Soundness: $x \notin L \Rightarrow \forall P_1, P_2, \ldots, P_{\text{poly}(|x|)}, Pr[\langle (P_1, P_2, \ldots, P_{\text{poly}(|x|)}), V \rangle_V (x) = 1] \leq \frac{1}{3}$

*Remark* C.9. The constant $C$ in the definition can be any constant. It won't affect the definition as long as it's a constant.

**Theorem C.10.** *MIP-Bounded = MIP*

Theorem C.10 establishes the efficacy of our framework. By building on the classical result of Babai et al. (1990), we derive Theorem C.11 that clearly illustrates the expressive power of our approach. Refer to Section C.3 for detailed proofs.

**Corollary C.11.** *NP ⊂ NEXP = MIP-Bounded*

The collollary indicates that if the verifier and the provers satisfy some condition, the LatentSeek framework is strong enough, and the independence assumption can be amended. Intuitively speaking, the independence between the provers can be controlled by the verifier, such as the verifier can ask for dependence between the generated tokens.

*There is a limitation in this theoretical analysis, which is why we present it in the appendix: it should be viewed only as an indication or heuristic guideline rather than a rigorous validation.* In particular, we do not account for the change through gradient updates. As a result, there is no guarantee that every token becomes reachable in each step, especially when each prover relies solely on the gradients from the language-model head for guidance.

**All in all, the theorem indicates that with a "strong" base model and a "well-aligned" reward model, the deficit introduced by the independence assumption can be effectively mitigated.**

## C.3 PROOFS OF THEOREM C.10 AND THEOREM C.11

**Theorem C.12.** *MIP-Bounded = MIP*

*Proof.* By the classical results of MIP = 2-MIP = NEXP Babai et al. (1990), it's sufficient to consider only two provers in the MIP class.

We will first prove that MIP $\subseteq$ MIP-Bounded.

For all $L \in$ MIP, $\forall x$, in the $j$-th turn, the proofs offered by the two provers are denoted as $a_{1j}, a_{2j}$, the verifier as $V$. According to the definition of MIP, $\exists$ polynomial function $\rho(\cdot)$ such that $|a_{ij}| \leq \rho(|x|))$, we set $\text{poly}(\cdot) = \frac{2\rho(\cdot)}{C}$, which is also polynomial. We design the verifier $V' = V$.

1. If $x \in L$, $\exists \text{poly}(|x|)$ and provers $P'_1, P'_2, \ldots, P'_{\text{poly}(|x|)}$ such that the accept rate is larger than $\frac{2}{3}$ in the following manner:

   The first $\frac{|a_{1j}|}{C}$ provers output $a_{1j}$ with each outputting $C$ bits, and the continuing $\frac{|a_{2j}|}{C}$ provers output $a_{2j}$ with each outputting $C$ bits. As $\text{poly}(|x|) = \frac{2\rho(|x|)}{C} \geq \frac{|a_{1j}|}{C} + \frac{|a_{2j}|}{C}$, we have enough provers to do this operation. Therefore, the outputs of $P'_1, P'_2, \ldots, P_{\text{poly}'(|x|)}$ is the same as the original two provers, as the original two provers satisfies that the accepts probability is greater than $\frac{2}{3}$, we have

$$Pr[\langle(P'_1, P'_2, \ldots, P'_{\text{poly}(|x|)}), V'\rangle_{V'}(x) = 1] \geq \frac{2}{3}$$

2. If $x \notin L$, $\forall P'_1, P'_2, \ldots, P_{\text{poly}'(|x|)}$, we denote their output at $j$-th turn as $a'_{1j}, a'_{2j}, a'_{3j}, \ldots, a'_{\text{poly}(|x|),j}$. Let the first prover output the concat output of odd provers, i.e. $a_{1j} = (a'_{1j}, a'_{2j}, \ldots, a'_{\lfloor \frac{\text{poly}(|x|)}{2} \rfloor,j})$, the second output the concat output of even provers, i.e. $a_{2j} = (a'_{\lceil \frac{\text{poly}(|x|)}{2} \rceil,j}, \ldots, a'_{\text{poly}(|x|),j})$. As no two provers can make the verifier accept at a probability larger than $\frac{1}{3}$, thus

$$Pr[\langle(P'_1, P'_2, \ldots, P'_{\text{poly}(|x|)}), V'\rangle_{V'}(x) = 1] \leq \frac{1}{3}$$

Therefore, $L \in$ MIP-Bounded, and thus MIP $\subseteq$ MIP-Bounded.

Next, we are going to prove MIP-Bounded $\subseteq$ MIP.

$\forall L \in$ MIP-Bounded, $\forall x$, in the $j$-th turn, the proofs offer by the bounded provers are denoted as: $a'_{1j}, a'_{2j}, a'_{3j}, \ldots, a'_{\text{poly}(|x|),j}$, the verifier as $V'$. We design $V$ as follows: for each concat string $s = (x, q_{11}, a_{11}, \ldots, )$, $V$ first truncates the first prover's answer to length of $\text{LIM}_1 := C \cdot \lfloor \frac{\text{poly}(|x|)}{2} \rfloor$ bits and the second prover's answer to length of $\text{LIM}_2 := C \cdot (\text{poly}(|x|) - \lceil \frac{\text{poly}(|x|)}{2} \rceil + 1)$ bits. The resulted string is denoted as $s' := (x, q_{11}, a_{11;0:\text{LIM}_1}, q_{21}, a_{21;0:\text{LIM}_2} \ldots)$, and output $V'(s')$.

1. If $x \in L$, similar as above, we let the first prover output the concat output of odd provers, i.e. $a_{1j} = (a'_{1j}, a'_{2j}, \ldots, a'_{\lfloor \frac{\text{poly}(|x|)}{2} \rfloor,j})$, the second output the concat output of even provers, i.e. $a_{2j} = (a'_{\lceil \frac{\text{poly}(|x|)}{2} \rceil,j}, \ldots, a'_{\text{poly}(|x|),j})$.. As $|a_{1j}| \leq \text{LIM}_1$, it won't be truncated and $|a_{2j}| \leq \text{LIM}_2$, it won't be truncated, thus $V$ acts exactly as $V'$, and therefore

$$Pr(\langle(P_1, P_2), V\rangle_V(x) = 1) \geq \frac{2}{3}$$

2. (Prove by contradiction) If $x \notin L$, suppose $\exists P_1, P_2$ such that:

$$Pr(\langle(P_1, P_2), V\rangle_V(x) = 1) > \frac{1}{3}$$

As $V$ first truncate answer, therefore, for $j$-turns, we can truncate $a_{1j}$ to $a_{1j;0:\text{LIM}_1}$ and $a_{2j}$ to $a_{2j;0:\text{LIM}_2}$, and follow the division process depicted in the first case of proving MIP $\subseteq$ MIP-Bounded to gain $P'_1, P'_2, P'_3, \ldots, P'_{\text{poly}(|x|)}$ provers. Thus:

$$Pr(\langle(P_1, P_2), V\rangle_V (x) = 1) = Pr[\langle(P'_1, P'_2, \ldots, P'_{\text{poly}(|x|)}), V\rangle_V (x) = 1]$$
$$= Pr[\langle(P'_1, P'_2, \ldots, P'_{\text{poly}(|x|)})', V'\rangle_{V'} (x) = 1]$$

As a result,

$$Pr[\langle(P'_1, P'_2, \ldots, P'_{\text{poly}(|x|)}), V'\rangle_{V'} (x) = 1] > \frac{1}{3}$$

, this contradicts to $Pr[\langle(P'_1, P'_2, \ldots, P'_{\text{poly}(|x|)}), V'\rangle_{V'} (x) = 1] \leq \frac{1}{3}$. Therefore, $\forall P_1, P_2$, we have:

$$Pr(\langle(P_1, P_2), V\rangle_V (x) = 1) \leq \frac{1}{3}$$

Hence, we have $L \in$ MIP, which means that MIP-Bounded $\subseteq$ MIP.

Concluding the above, we have MIP-Bounded = MIP. $\square$

**Theorem C.13** (Babai et al. (1990)). *MIP = 2-MIP = NEXP*

*Remark* C.14. NEXP is the complexity class that a non-deterministic Turing Machine can solve in exponential time, which means that NP $\subset$ NEXP.

Based on this theorem, we can easily derive Theorem C.11.

**Corollary C.15.** *NP $\subset$ NEXP = MIP-Bounded*

# D  DERIVATION OF POLICY GRADIENT

Our target is to derive Equation (7):

$$[\nabla_{\mathbf{z}}\mathcal{J}(\mathbf{z})]_t = \mathbb{E}_{\mathbf{x} \sim \pi(\mathbf{x}|\mathbf{z}, \mathbf{c})} [R(\mathbf{x}, \mathbf{c})\nabla_{z_t} \log \pi(x_t \mid z_t)]$$

Starting from the objective:

$$\mathcal{J}(\mathbf{z}) := \mathbb{E}_{\mathbf{x} \sim \pi(\mathbf{x}|\mathbf{z}, \mathbf{c})} [R(\mathbf{x}, \mathbf{c})]$$

By taking gradient of $\mathbf{z}$, we have:

$$\nabla_{\mathbf{z}}\mathcal{J}(\mathbf{z}) = \nabla_{\mathbf{z}} \int_{\mathbf{x}} R(\mathbf{x}, \mathbf{c})\pi(\mathbf{x} \mid \mathbf{z}, \mathbf{c})d\mathbf{x} = \int_{\mathbf{x}} R(\mathbf{x}, \mathbf{c})\nabla_{\mathbf{z}}\pi(\mathbf{x} \mid \mathbf{z}, \mathbf{c})d\mathbf{x}$$

Leveraging $\nabla_{\mathbf{z}} \log \pi(\mathbf{x} \mid \mathbf{z}, \mathbf{c}) = \frac{1}{\pi(\mathbf{x}|\mathbf{z}, \mathbf{c})}\nabla_{\mathbf{z}}\pi(\mathbf{x} \mid \mathbf{z}, \mathbf{c})$:

$$\nabla_{\mathbf{z}}\mathcal{J}(\mathbf{z}) = \int_{\mathbf{x}} R(\mathbf{x}, \mathbf{c})\pi(\mathbf{x} \mid \mathbf{z}, \mathbf{c})\nabla_{\mathbf{z}} \log \pi(\mathbf{x} \mid \mathbf{z}, \mathbf{c})d\mathbf{x}$$

According to $\pi(\mathbf{x} \mid \mathbf{z}, \mathbf{c}) = \prod_{t=1}^{N} \pi(x_t \mid z_t) \prod_{t=N+1}^{T} \pi(x_t|x_{<t}, \mathbf{c})$, we have:

$$\nabla_{\mathbf{z}}\mathcal{J}(\mathbf{z}) = \int_{\mathbf{x}} R(\mathbf{x}, \mathbf{c})\pi(\mathbf{x} \mid \mathbf{z}, \mathbf{c})\nabla_{\mathbf{z}}(\sum_{t=1}^{N} \log \pi(x_t \mid z_t) + \sum_{t=N+1}^{T} \log \pi(x_t|x_{<t}))d\mathbf{x}$$

As the second term doesn't have $\mathbf{z}$, thus we have:

$$\nabla_{\mathbf{z}}\mathcal{J}(\mathbf{z}) = \int_{\mathbf{x}} R(\mathbf{x}, \mathbf{c})\pi(\mathbf{x} \mid \mathbf{z}, \mathbf{c})\nabla_{\mathbf{z}}(\sum_{t=1}^{N} \log \pi(x_t \mid z_t))d\mathbf{x} = \mathbb{E}_{\mathbf{x} \sim \pi(\mathbf{x}|\mathbf{z}, \mathbf{c})}[R(\mathbf{x}, \mathbf{c})\nabla_{\mathbf{z}}(\sum_{t=1}^{N} \log \pi(x_t|z_t))]$$

Therefore, the $t$-th term is:

$$[\nabla_{\mathbf{z}}\mathcal{J}(\mathbf{z})]_t = \mathbb{E}_{\mathbf{x} \sim \pi(\mathbf{x}|\mathbf{z}, \mathbf{c})}[R(\mathbf{x}, \mathbf{c})\nabla_{z_t} \log \pi(x_t|z_t)],$$

which is exactly what we want.

# E ADDITIONAL EXPERIMENTAL RESULTS

## E.1 LATENTSEEK VS BON (N=10)

In Table 1, we show that Best-of-N (BoN) is outperformed by LATENTSEEK, even though BoN **consumes a comparable compute consumption (See Figure 4)**. One might also ask how LATENTSEEK compares to BoN when $N = 10$; in this setting, BoN requires at least five times more sequence-level computation than LATENTSEEK. As reported in Table 10, LATENTSEEK remains superior on all datasets and all backbone models.

Table 10: Additional Comparison Between LATENTSEEK and BoN (N=10) with Prompt 2.

| Model
Methods | Qwen2
7B | 1.5B | Qwen2.5
7B | 14B | LLaMA3.1
8B | Avg |
|---|---|---|---|---|---|---|
| *GSM8K* | | | | | | |
| BoN | 72.10 | 7.73 | 80.06 | 92.19 | 81.58 | 66.73 |
| LATENTSEEK | $80.21_{+8.11}$ | $44.20_{+36.47}$ | $85.06_{+5.00}$ | $92.72_{+0.53}$ | $83.70_{+2.12}$ | $77.18_{+10.45}$ |
| *MATH-500* | | | | | | |
| BoN | 42.40 | 28.80 | 56.40 | 63.20 | 45.60 | 47.28 |
| LATENTSEEK | $44.80_{+2.40}$ | $32.20_{+3.40}$ | $57.60_{+1.20}$ | $71.00_{+7.80}$ | $47.00_{+1.40}$ | $50.52_{+3.24}$ |
| *AIME2024* | | | | | | |
| BoN | 3.33 | 0.00 | 6.67 | 10.00 | 3.33 | 4.66 |
| LATENTSEEK | $3.33_{+0.00}$ | $3.33_{+3.33}$ | $13.33_{+6.66}$ | $10.00_{+0.00}$ | $6.67_{+3.33}$ | $7.33_{+2.67}$ |

## E.2 QWEN3 AIME PERFORMANCE

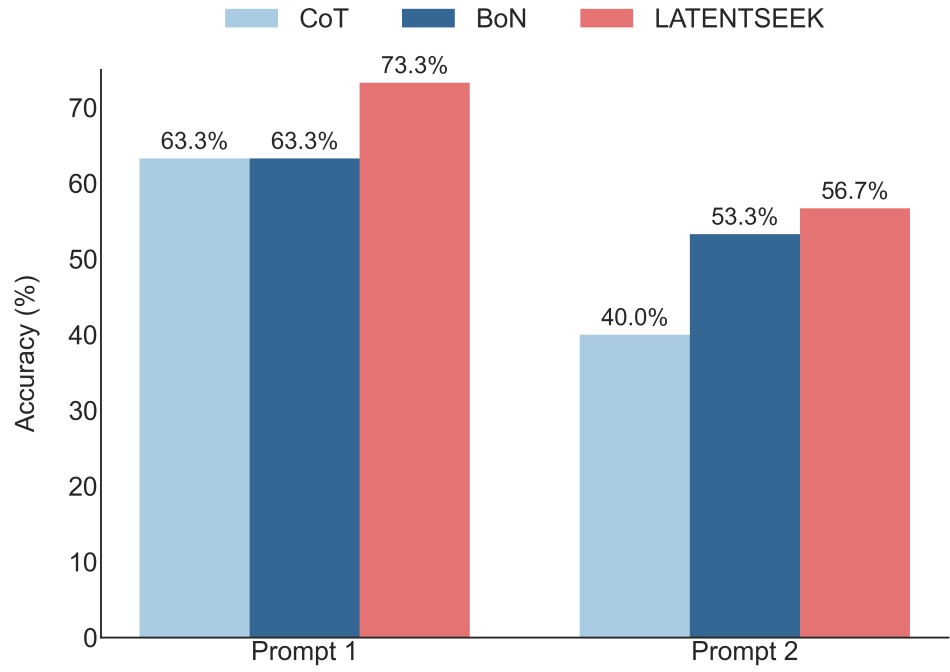

Figure 3: Performance comparison of the Qwen3-4B-Instruct model on the AIME2024 dataset. The chart illustrates the accuracy of LATENTSEEK against the Chain-of-Thought (CoT) and Best-of-N (BoN) baselines across two distinct prompt formats.

This section provides a detailed study on the performance of the Qwen3-4B-Instruct-2507 model on the challenging AIME2024 dataset. Figure 3 illustrates the accuracy of LATENTSEEK compared against the Chain-of-Thought (CoT) and Best-of-N (BoN) baselines under two prompt formats.

As shown in the figure, under Prompt 1, while the BoN strategy fails to improve upon the CoT baseline (both at 63.3% accuracy score), LATENTSEEK delivers a significant **10 points uplift** to 73.3%. This advantage holds under Prompt 2, where our method's accuracy of 56.7% surpasses CoT by **16.7 points** and BoN by **3.4 points**. Taken together, these findings demonstrate that LATENTSEEK **consistently improves upon strong baselines** and is valuable in scenarios where conventional search methods are insufficient.

### E.3 TOKEN EFFICIENCY COMPARISON WITH BoN BASELINE

This section provides a quantitative analysis of the token efficiency of LATENTSEEK, comparing it against the Best-of-N (BoN) baseline reported in Table 1. The analysis measures the average number of tokens consumed per problem instance under the JSON prompt format (Prompt 2) on the GSM8K and MATH-500 datasets.

As shown in Figure 4, LATENTSEEK achieves higher token efficiency than BoN in all of cases across different model families and sizes, while attaining superior reasoning accuracy as reported in Table 1. These results demonstrates that LATENTSEEK can enhance performance without incurring additional token overhead for all models, highlighting the efficiency of latent space optimization.

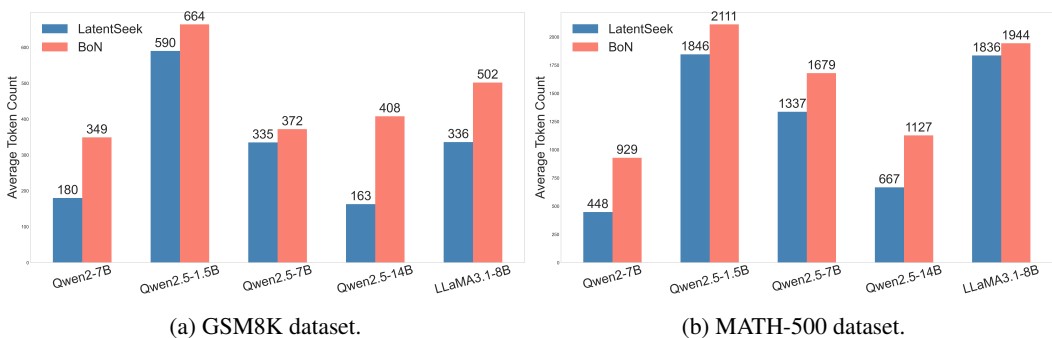

(a) GSM8K dataset.        (b) MATH-500 dataset.

Figure 4: Comparison of average tokens consumed per problem between LATENTSEEK and the Best-of-N (BoN) baseline when using Prompt 2. The token consumption is comparable for the two methods, with BoN consuming slightly more tokens in total. For experimental results where BoN consumes extremely more calculation, please refer to Table 10.

### E.4 GREEDY DECODING VS SAMPLING

Our experiments show that greedy decoding closely approximates categorical sampling while being a better choice for reducing stochastic error. As detailed in Table 11, greedy decoding performs slightly better than the sampling method. We hypothesize that this is because sampling can amplify the approximation error when replacing an expected mean with an empirical one. Therefore, we use greedy decoding for our main experiments.

Table 11: LATENTSEEK: Greedy Decoding vs Sampling (Temperature 0.7) on the MATH-500 dataset.

| Model Methods | Qwen2 7B | Qwen2.5 1.5B | 7B | 14B | LLaMA3.1 8B |
|---|---|---|---|---|---|
| *Prompt 1* | | | | | |
| Sampling | 56.80 | 50.00 | 74.00 | 74.00 | 49.60 |
| Greedy Decoding | 57.40 | 55.60 | 75.60 | 80.00 | 54.60 |
| *Prompt 2* | | | | | |
| Sampling | 44.40 | 22.40 | 55.80 | 68.40 | 45.40 |
| Greedy Decoding | 44.80 | 34.40 | 57.60 | 71.00 | 47.00 |

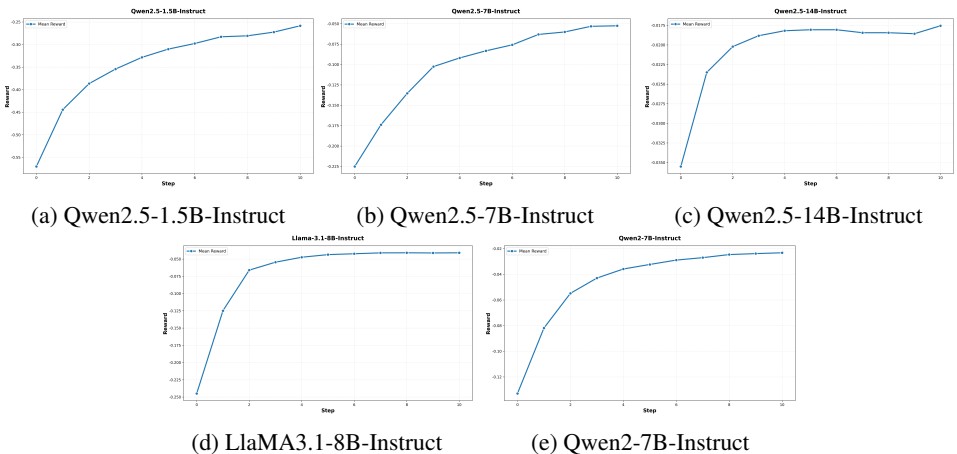

(a) Qwen2.5-1.5B-Instruct     (b) Qwen2.5-7B-Instruct     (c) Qwen2.5-14B-Instruct

(d) LlaMA3.1-8B-Instruct     (e) Qwen2-7B-Instruct

Figure 5: Trends of reward on GSM8K dataset. Generation uses the prompt 2 (the Json prompt).

### E.5   LIMITATIONS OF OUTCOME REWARD MODEL

The performance is described in Liu et al. (2024b), with results presented in Table 12. Although this model achieves an average score of 64.79—representing an improvement of 3.23 points over the Chain-of-Thought (CoT) method—it remains significantly inferior to the performance attained using a self-rewarding mechanism, which reaches 77.18. These results highlight the limitations of current outcome-based reward models in scenarios lacking ground-truth data, underscoring the need for the development of more robust and effective outcome reward models.

Table 12: Math Reward Model Accuracy results on GSM8k Cobbe et al. (2021), MATH-500 Hendrycks et al. (2021), and AIME2024 datasets. The best performances are highlighted in bold, and the second-best performances are underlined.

| Model Methods | Qwen2 7B | Qwen2.5 1.5B | Qwen2.5 7B | Qwen2.5 14B | LLaMA3.1 8B | Avg |
|---|---|---|---|---|---|---|
| *GSM8k: Prompt 2* | | | | | | |
| CoT | 65.20 | 15.31 | 66.41 | 91.81 | 69.07 | 61.56 |
| LATENTSEEK (*Math*) | 61.18 | 28.28 | 66.64 | 91.05 | 76.80 | 64.79 |
| **LATENTSEEK** (*Self*) | **80.21** | **44.20** | **85.06** | **92.72** | **83.70** | **77.18** |

### E.6   TRENDS OF REWARD

We plot the reward trends in Figure 5, which empirically illustrates how the update dynamics guide the model from a low-reward region toward a higher-reward region. The reward curves are obtained by running LATENTSEEK on the GSM8K dataset, with generation constrained by a JSON-formatted prompt.

### E.7   COMPARISON OF OPTIMIZING INITIAL-STAGE LATENT TOKENS AND OPTIMIZING MIDDLE-STAGE LATENT TOKENS

One may wonder how performance changes when we instead optimize **middle-stage** tokens. Formally, this corresponds to the factorization

$$\pi(\mathbf{x} \mid \mathbf{z}, \mathbf{c}) = \prod_{t=1}^{N_1} \pi(x_t \mid \mathbf{x}_{<t}, \mathbf{c}) \prod_{t=N_1+1}^{N} \pi_{\text{LM-head}}(x_t \mid z_t) \prod_{t=N+1}^{T} \pi(x_t \mid \mathbf{x}_{<t}, \mathbf{c}).$$

Intuitively, because the prefix tokens $x_1$ to $x_{N_1}$ do not depend on the optimized middle-stage tokens $\mathbf{x}_{N_1+1:N}$, the model may be unable to fully exploit the improved latent representations, potentially

Table 13: Comparison with Optimizing the Middle-Stage Tokens on GSM8K. The best preformances are highlighted in bold.

| Model | Qwen2 7B | Qwen2.5 1.5B | 7B | 14B | LLaMA3.1 8B | Avg |
|---|---|---|---|---|---|---|
| *GSM8K: Prompt 1* | | | | | | |
| CoT | 68.01 | 68.08 | 88.86 | 92.03 | 50.19 | 73.43 |
| LATENTSEEK (middle) | 74.68 | 69.07 | 89.31 | 92.49 | 65.21 | 78.15 |
| **LATENTSEEK** | **84.38** | **70.89** | **90.14** | **92.49** | **78.54** | **83.29** |
| *GSM8K: Prompt 2* | | | | | | |
| CoT | 65.20 | 15.31 | 66.41 | 91.81 | 69.07 | 61.56 |
| LATENTSEEK (middle) | 70.73 | 22.44 | 72.93 | 92.49 | 77.26 | 67.17 |
| **LATENTSEEK** | **80.21** | **44.20** | **85.06** | **92.72** | **83.70** | **77.18** |

causing performance degradation. To test this hypothesis, we conduct an additional experiment where we optimize tokens starting from (40%) into the sequence. The results, shown in Table 13, confirm our intuition: although optimizing middle-stage tokens yields a 5.16-point improvement over CoT on average, it still underperforms the version that optimizes the **initial-stage** tokens by 7.58 points. This gap supports our hypothesis that prefix independence limits the effectiveness of optimizing middle-stage tokens.

### E.8    PERFORMANCE VS FRACTION RATIO

Figure 6 illustrates the performance trends of various instruction-tuned language models across different fraction ratios, which range from 0.1 to 0.8. Here, we also experimented with another model: *Mistral-7B-Instruct* (Jiang et al., 2023). Performance is reported as a percentage and plotted on the y-axis, while the fraction ratio is shown on the x-axis.

- *Qwen2.5-14B-Instruct* exhibits a relatively stable performance curve. It shows a slight increase in performance from 0.1 to 0.6, peaking around the 0.6 fraction ratio, followed by a minor decrease at 0.8. Overall, this model maintains consistently high performance above 91% across all fraction ratios.

- *Qwen2.5-7B-Instruct* demonstrates a mild upward trend from 0.1 to 0.2, followed by a steady decrease as the fraction ratio increases beyond 0.2, indicating reduced effectiveness at higher ratios.

- *Qwen2-7B-Instruct* shows a consistent downward trend across the full range of fraction ratios, suggesting that its performance deteriorates steadily with increasing input fraction.

- *LLaMA3.1-8B-Instruct* remains relatively stable at first but shows a slight decreasing trend overall, with performance gently declining from 0.2 onwards.

- *Qwen2.5-1.5B-Instruct* follows a sharp non-monotonic trend. It increases markedly from 0.1 to peak at 0.2, then drops rapidly as the fraction ratio increases further, reaching its lowest performance at 0.8.

- *Mistral-7B-Instruct* exhibits a subtle peak at 0.2, followed by a gradual and modest decrease in performance as the fraction ratio increases, maintaining relatively stable performance throughout.

In summary, the Qwen2.5-14B-Instruct model shows a slightly rising then declining trend but remains highly stable and strong overall. Smaller models, especially Qwen2.5-1.5B-Instruct, are more sensitive to increases in fraction ratio, with noticeable performance degradation at higher values. Mid-sized models like Mistral-7B-Instruct and LLaMA3.1-8B-Instruct exhibit relatively mild downward trends, indicating moderate robustness.

## F    EXPERIMENTAL DETAILS

In this section, we provide a detailed description of the experimental setup.

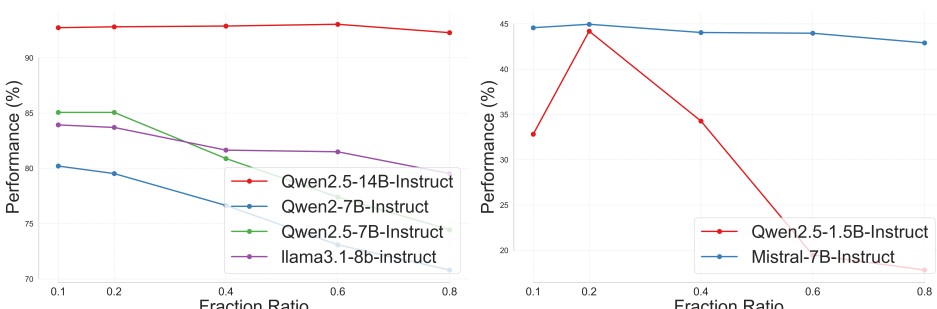

Figure 6: Performance vs Fraction Ratio

### F.1 PROMPT DESIGNATION

The corresponding prompts are shown in the following tables.

**Prompt.** The prompt of CoT is as follows.

- Prompt 1: Table 15 and Table 16.
- Prompt 2: Table 14.

**Reward Prompt.** The prompts are listed in Tables 23 to 26.

### F.2 BACKBONE

We list all model backbone as follows:

- Qwen2-7B-Instruct Yang et al. (2024): A powerful model designed for instruction-based tasks, leveraging the 7B parameter version of Qwen2. (accessible at `https://huggingface.co/Qwen/Qwen2-7B-Instruct`)
- Qwen2.5-1.5B-Instruct Team (2024a): A compact yet efficient model designed for task-specific instructions, based on the 1.5B-parameter Qwen2.5. (accessible at `https://huggingface.co/Qwen/Qwen2.5-1.5B-Instruct`)
- Qwen2.5-7B-Instruct Team (2024a): A middle-tier model based on 7B-parameter Qwen2.5, optimized for handling various instructions. (accessible at `https://huggingface.co/Qwen/Qwen2.5-7B-Instruct`)
- Qwen2.5-14B-Instruct Team (2024a): A robust, large-scale model built on the 14B-parameter Qwen2.5, excelling in complex instruction-based tasks. (accessible at `https://huggingface.co/Qwen/Qwen2.5-14B-Instruct`)
- LLaMA3.1-8B-Instruct Team (2024b): LLaMA's 8B parameter version designed for better instruction-following capabilities. (accessible at `https://huggingface.co/meta-llama/Llama-3.1-8B-Instruct`)

### F.3 BASELINES

We describe all baselines in experiments as follows:

- Chain-of-Thought (CoT): CoT refers to a structured, sequential approach to problem-solving, wherein complex tasks are decomposed into intermediate steps, each explicitly articulated to facilitate logical progression toward a solution. In our experiments, CoT follows the prompt listed in Section F.1.
- Few-Shot CoT: Few-Shot Chain-of-Thought (CoT) prompting is a technique in natural language processing that enhances model performance on complex reasoning tasks by providing a small number of illustrative examples with step-by-step reasoning, enabling the model to generalize and produce structured, logical outputs. In our experiment, we adopt the result from Lambert et al. (2025).

Table 14: Prompt 1 for GSM8k Cobbe et al. (2021), MATH-500 Hendrycks et al. (2021) and AIME2024.

```
{"role": "system", "content": "Please reason step by step, and
put your final answer within \boxed\{\}." },
\{"role": "user", "content": "{q}"\}
```

- Best-of-$N$ (BoN): We draw $N$ independent candidates from the model, score each with a predefined objective, and select the candidate with the highest reward. In our experiments, we set $N = 3$ so that the total computation is comparable to—while exceeding—that of LATENTSEEK. The BoN reward is computed under the same self-reward mechanism as that of LATENTSEEK.

- Self-Consistency: We draw $N$ independent candidates from the model, and select the candidate that appears with the highest frequency. In our experiments, we set $N = 3$ so that the total computation is comparable to—while exceeding—that of LATENTSEEK and align with that of BoN.

- Self-Rewarding Yuan et al. (2025): This leverages intrinsic feedback mechanisms to iteratively enhance model performance without reliance on external reward signals.

- ScPO Prasad et al. (2024): A framework for self-consistency preference optimization, rigorously formalizing the alignment of decision-making processes with logically coherent and preference-driven outcomes in complex systems.

- CoH Liu et al. (2023a): A framework which systematically aligns language models with human feedback through a structured, iterative process, enhancing their performance in complex reasoning tasks.

- Genius Xu et al. (2025): A purely unsupervised self-training framework designed to enhance advanced reasoning capabilities in artificial intelligence systems, offering generalizable performance across diverse tasks without reliance on labeled data.

- SimpleRL-Zoo Zeng et al. (2025): A reinforcement learning framework designed to enhance sample efficiency and performance stability in complex decision-making environments through simplified algorithmic structures and adaptive exploration strategies.

- GRPO Shao et al. (2024): Group Relative Policy Optimization (GRPO), introduced in the DeepSeekMath framework, is a novel reinforcement learning algorithm that enhances mathematical reasoning in large language models by optimizing policy updates through group-based reward comparisons, significantly reducing memory consumption compared to traditional Proximal Policy Optimization (PPO). In our experiment, we adopt the result from https://www.perplexity.ai/hub/blog/rl-training-for-math-reasoning?utm_source=chatgpt.com.

- SPIN Chen et al. (2024a): A self-play fine-tuning methodology that significantly enhances the performance of weaker language models, transforming them into robust and highly capable systems competitive with stronger counterparts.

- iCoT Deng et al. (2024): a transition from explicit Chain-of-Thought (CoT) reasoning to implicit CoT internalization, proposing a step-by-step learning framework to enhance logical reasoning capabilities in artificial intelligence systems.

- Supervised Fine-Tuning (SFT) was conducted using the LLaMA-Factory framework Zheng et al. (2024). All models were trained with a learning rate of $1 \times 10^{-5}$, employing a cosine learning rate scheduler, a warmup ratio of 0.1, and the bfloat16 (bf16) data type.

## F.4 HYPERPARAMETER TUNING

To determine the best choices for $\rho$ and the learning-rate hyperparameters, we perform a grid search on a held-out subset of the training data. For GSM8K, for example, we randomly sample 500 training instances and use this subset exclusively for hyperparameter tuning.

Table 15: Prompt 2 for GSM8k Cobbe et al. (2021).

```
\{"role": "system", "content": "You are a precise math question
solver. Solve this math problem." \},
\{"role": "user", "content": "QUESTION: {q} Let's think step by
step. Please provide your thought process and your final answer
separately and response in json format containing the keys
thought process and final answer. For example your response
should be \{'thought process': 'your thought process', 'final
answer': 'your final answer'\}. Note that the final answer
should be pure numbers, not the calculation formulas, and
without any units or explanation!!!"\}
```

Table 16: Prompt 2 for MATH-500 Hendrycks et al. (2021) and AIME2024.

```
\{"role": "system", "content": "You are a precise math question
solver. Solve this math problem." \},
\{"role": "user", "content": "QUESTION: {q} Let's think step by
step. Please provide your thought process and your final answer
separately and response in json format containing the keys
thought process and final answer. For example your response
should be \{'thought process': 'your thought process', 'final
answer': 'your final answer'\}." \}
```

### F.5 GSM8K

We provide details for GSM8K experiments as follows:

**Dataset.** The GSM8K dataset Cobbe et al. (2021), a comprehensive collection of mathematical reasoning problems, serves as a benchmark for evaluating the problem-solving capabilities of language models. Developed by OpenAI and accessible via the Hugging Face repository at https://huggingface.co/datasets/openai/gsm8k, GSM8K comprises 8,500 meticulously curated, high-quality math problems that span a diverse range of topics, including arithmetic, algebra, and word problems. These problems are specifically designed to assess a model's ability to perform multi-step reasoning, interpret natural language descriptions of mathematical scenarios, and derive accurate solutions. The dataset is partitioned into a training set of 7,473 examples and a test set of 1,319 examples, enabling robust model training and evaluation. For the purposes of this study, we exclusively utilize the test set to evaluate model performance, ensuring a standardized and unbiased assessment of mathematical reasoning proficiency.

**Experimental Details.** For all backbones and both prompts, we use greedy decoding for inference. For the hyperparameters of LATENTSEEK (Self) and LATENTSEEK (PSRM), please refer to Table 17 and Table 18, respectively. The mathematical reasoning prompts we employed in the self-reward mechanism evaluate answers across four dimensions: correctness of the final answer, accuracy of problem comprehension, correctness of numerical calculations, and provision of a clear answer, weighted at 1:1:2:2, with the final score normalized to the range [-1, 0]. The detailed specifications of these four evaluation prompts are provided in Tables 23 to 26.The prompt structure is consistently applied across all backbones and both prompts. Notably, for Llama-3.1-8B-Instruct, Qwen2.5-1.5B-Instruct and Qwen2-7B-Instruct in LATENTSEEK (Self) experiment using Prompt 1, a new format reward following the methodology of DeepSeek-R1 is incorporated, with weights of 3, 2, and 2 assigned to format-based criteria, respectively.

**Scaling.** We offer more scaling figures in Figure 7.

### F.6 MATH-500

We provide details for MATH-500 experiments as follows:

**Dataset.** The MATH-500 dataset, a curated subset of the MATH benchmark, serves as a robust resource for evaluating the mathematical reasoning capabilities of machine learning mod-

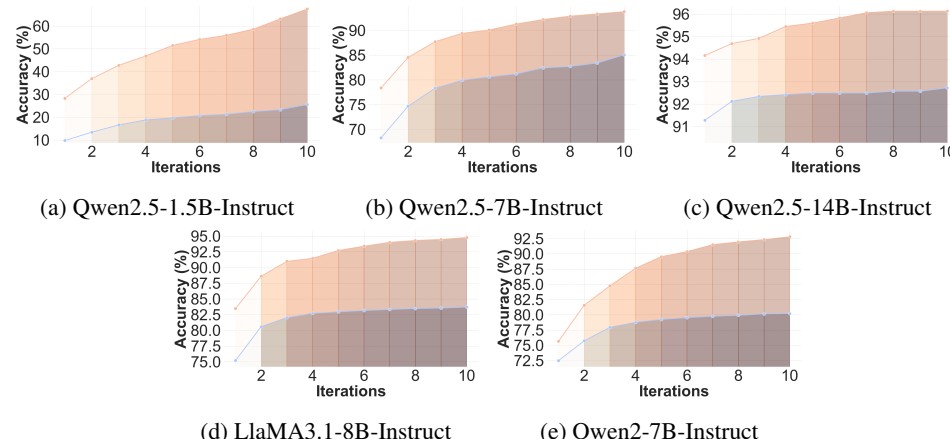

Figure 7: GSM8kCobbe et al. (2021) Prompt 2 Accuracy changes with respect to the increasing number of iterations. Orange: Perfect Sharp Reward Model. Blue: Self Reward Model.

Table 17: LATENTSEEK (Self) Hyperparameters on GSM8K. lr: learning rate. $\rho$: fraction ratio

| methods | model | max len | prompt idx | #GPU | lr | optimizer | $\rho$ | dtype | Max Step |
|---|---|---|---|---|---|---|---|---|---|
| LATENTSEEK (Self) | Qwen2-7B-Instruct | 1024 | 1 | 1 A100 | 0.03 | Adam | 0.2 | bf16 | 10 |
| LATENTSEEK (Self) | Qwen2-7B-Instruct | 1024 | 2 | 1 L40 | 0.03 | Adam | 0.2 | bf16 | 10 |
| LATENTSEEK (Self) | Qwen2.5-1.5B-Instruct | 1024 | 1 | 1 3090 | 0.03 | Adam | 0.05 | bf16 | 10 |
| LATENTSEEK (Self) | Qwen2.5-1.5B-Instruct | 1024 | 2 | 1 3090 | 10 | Adam | 0.2 | bf16 | 10 |
| LATENTSEEK (Self) | Qwen2.5-7B-Instruct | 1024 | 1 | 1 L40 | 0.05 | Adam | 0.2 | bf16 | 10 |
| LATENTSEEK (Self) | Qwen2.5-7B-Instruct | 1024 | 2 | 1 L40 | 0.05 | Adam | 0.2 | bf16 | 10 |
| LATENTSEEK (Self) | Qwen2.5-14B-Instruct | 1024 | 1 | 1 L40 | 0.03 | Adam | 0.1 | bf16 | 10 |
| LATENTSEEK (Self) | Qwen2.5-14B-Instruct | 1024 | 2 | 1 L40 | 0.03 | Adam | 0.1 | bf16 | 10 |
| LATENTSEEK (Self) | Llama-3.1-8B-Instruct | 1024 | 1 | 1 L40 | 0.03 | Adam | 0.2 | bf16 | 10 |
| LATENTSEEK (Self) | Llama-3.1-8B-Instruct | 1024 | 2 | 1 L40 | 0.03 | Adam | 0.2 | bf16 | 10 |

Table 18: LATENTSEEK (PSRM) Hyperparameters on GSM8K. lr: learning rate. $\rho$: fraction ratio

| methods | model | max len | prompt idx | #GPU | lr | optimizer | $\rho$ | dtype | Max Step |
|---|---|---|---|---|---|---|---|---|---|
| LATENTSEEK (PSRM) | Qwen2-7B-Instruct | 1024 | 1 | 1 A100 | 0.03 | Adam | 0.2 | bf16 | 10 |
| LATENTSEEK (PSRM) | Qwen2-7B-Instruct | 1024 | 2 | 1 4090 | 0.03 | Adam | 0.2 | bf16 | 10 |
| LATENTSEEK (PSRM) | Qwen2.5-1.5B-Instruct | 1024 | 1 | 1 3090 | 0.03 | Adam | 0.2 | bf16 | 10 |
| LATENTSEEK (PSRM) | Qwen2.5-1.5B-Instruct | 1024 | 2 | 1 3090 | 0.03 | Adam | 0.2 | bf16 | 10 |
| LATENTSEEK (PSRM) | Qwen2.5-7B-Instruct | 1024 | 1 | 1 A100 | 0.05 | Adam | 0.2 | bf16 | 10 |
| LATENTSEEK (PSRM) | Qwen2.5-7B-Instruct | 1024 | 2 | 1 L40 | 0.05 | Adam | 0.2 | bf16 | 10 |
| LATENTSEEK (PSRM) | Qwen2.5-14B-Instruct | 1024 | 1 | 1 A100 | 0.03 | Adam | 0.1 | bf16 | 10 |
| LATENTSEEK (PSRM) | Qwen2.5-14B-Instruct | 1024 | 2 | 1 A100 | 0.03 | Adam | 0.1 | bf16 | 10 |
| LATENTSEEK (PSRM) | Llama-3.1-8B-Instruct | 1024 | 1 | 1 L40 | 0.03 | Adam | 0.2 | bf16 | 10 |
| LATENTSEEK (PSRM) | Llama-3.1-8B-Instruct | 1024 | 2 | 1 L40 | 0.03 | Adam | 0.2 | bf16 | 10 |

els. Sourced from the HuggingFace repository at `https://huggingface.co/datasets/HuggingFaceH4/MATH-500`, it comprises 500 problems derived from the original MATH benchmark developed by OpenAI (Lightman et al., 2023). Encompassing a diverse array of mathematical topics and varying difficulty levels, MATH-500 provides a comprehensive and challenging testbed for assessing model performance in mathematical problem-solving.

**Experimental Details.** For all backbones and both prompts, we use greedy decoding for inference. For the hyperparameters of LATENTSEEK (Self) and LATENTSEEK (PSRM), please refer to Table 19 and Table 20, respectively. The mathematical reasoning prompts we employed in the self-reward mechanism evaluate answers across four dimensions: correctness of the final answer, accuracy of problem comprehension, correctness of numerical calculations, and provision of a clear answer, weighted at 1:1:2:2, with the final score normalized to the range [-1, 0]. The prompt structure is consistently applied across all backbones and both prompts. Notably, for Qwen2-7B-Instruct and Llama-3.1-8B-Instruct in LATENTSEEK (Self) experiment using prompt 1, the weight ratios are adjusted to 1:1:1:2. Additionally, for Qwen2-7B-Instruct, Qwen2.5-1.5B-Instruct, Qwen2.5-7B-Instruct and Llama-3.1-8B-Instruct in LATENTSEEK (Self) experiment using Prompt 1, a new format

Table 19: LATENTSEEK (Self) Hyperparameters on MATH-500. lr: learning rate. $\rho$: fraction ratio

| methods | model | max len | prompt idx | #GPU | lr | optimizer | $\rho$ | dtype | Max Step |
|---|---|---|---|---|---|---|---|---|---|
| LATENTSEEK (Self) | Qwen2-7B-Instruct | 1024 | 1 | 1 3090 | 0.03 | Adam | 0.1 | bf16 | 10 |
| LATENTSEEK (Self) | Qwen2-7B-Instruct | 1024 | 2 | 1 3090 | 0.03 | Adam | 0.1 | bf16 | 10 |
| LATENTSEEK (Self) | Qwen2.5-1.5B-Instruct | 1024 | 1 | 1 A100 | 0.03 | Adam | 0.1 | bf16 | 10 |
| LATENTSEEK (Self) | Qwen2.5-1.5B-Instruct | 1024 | 2 | 1 3090 | 0.03 | Adam | 0.05 | bf16 | 10 |
| LATENTSEEK (Self) | Qwen2.5-7B-Instruct | 1024 | 1 | 1 3090 | 0.03 | Adam | 0.1 | bf16 | 10 |
| LATENTSEEK (Self) | Qwen2.5-7B-Instruct | 1024 | 2 | 1 3090 | 0.03 | Adam | 0.1 | bf16 | 10 |
| LATENTSEEK (Self) | Qwen2.5-14B-Instruct | 1024 | 1 | 1 A100 | 0.03 | Adam | 0.1 | bf16 | 10 |
| LATENTSEEK (Self) | Qwen2.5-14B-Instruct | 1024 | 2 | 1 A100 | 0.05 | Adam | 0.1 | bf16 | 10 |
| LATENTSEEK (Self) | Llama-3.1-8B-Instruct | 1024 | 1 | 1 A100 | 0.03 | Adam | 0.1 | bf16 | 10 |
| LATENTSEEK (Self) | Llama-3.1-8B-Instruct | 1024 | 2 | 1 A100 | 0.03 | Adam | 0.1 | bf16 | 10 |

Table 20: LATENTSEEK (PSRM) Hyperparameters on MATH-500. lr: learning rate. $\rho$: fraction ratio

| methods | model | max len | prompt idx | #GPU | lr | optimizer | $\rho$ | dtype | Max Step |
|---|---|---|---|---|---|---|---|---|---|
| LATENTSEEK (PSRM) | Qwen2-7B-Instruct | 1024 | 1 | 1 3090 | 0.03 | Adam | 0.1 | bf16 | 10 |
| LATENTSEEK (PSRM) | Qwen2-7B-Instruct | 1024 | 2 | 1 3090 | 0.03 | Adam | 0.1 | bf16 | 10 |
| LATENTSEEK (PSRM) | Qwen2.5-1.5B-Instruct | 1024 | 1 | 1 A100 | 0.03 | Adam | 0.1 | bf16 | 10 |
| LATENTSEEK (PSRM) | Qwen2.5-1.5B-Instruct | 1024 | 2 | 1 A100 | 0.03 | Adam | 0.1 | bf16 | 10 |
| LATENTSEEK (PSRM) | Qwen2.5-7B-Instruct | 1024 | 1 | 1 3090 | 0.03 | Adam | 0.1 | bf16 | 10 |
| LATENTSEEK (PSRM) | Qwen2.5-7B-Instruct | 1024 | 2 | 1 3090 | 0.03 | Adam | 0.1 | bf16 | 10 |
| LATENTSEEK (PSRM) | Qwen2.5-14B-Instruct | 1024 | 1 | 1 A100 | 0.03 | Adam | 0.1 | bf16 | 10 |
| LATENTSEEK (PSRM) | Qwen2.5-14B-Instruct | 1024 | 2 | 1 A100 | 0.03 | Adam | 0.1 | bf16 | 10 |
| LATENTSEEK (PSRM) | Llama-3.1-8B-Instruct | 1024 | 1 | 1 A100 | 0.03 | Adam | 0.1 | bf16 | 10 |
| LATENTSEEK (PSRM) | Llama-3.1-8B-Instruct | 1024 | 2 | 1 A100 | 0.03 | Adam | 0.1 | bf16 | 10 |

reward following the methodology of DeepSeek-R1 is incorporated, with weight of 2 assigned to format-based criteria.

## F.7 AIME2024

We provide details for AIME2024 experiments as follows:

**Dataset.** The American Invitational Mathematics Examination (AIME) is a prestigious competition designed to challenge high-achieving high school students with complex mathematical problems, requiring advanced problem-solving and reasoning skills. The AIME2024 dataset, as introduced in this context, serves as a valuable resource for evaluating the capabilities of language models in tackling such sophisticated mathematical tasks. Sourced from the Huggingface repository Maxwell-Jia/AIME_2024 (accessible at https://huggingface.co/datasets/Maxwell-Jia/AIME_2024), the AIME2024 dataset comprises 30 meticulously curated problems. Although modest in quantity, each problem is deliberately designed to reflect the style, rigor, and difficulty of the AIME, thereby providing a robust benchmark for assessing advanced mathematical reasoning in computational models.

**Experimental Details.** For all backbones and both prompts, we use greedy decoding for inference. For the hyperparameters of LATENTSEEK (Self) and LATENTSEEK (PSRM), please refer to Table 21 and Table 22, respectively. The mathematical reasoning prompts we employed in the self-reward mechanism evaluate answers across four dimensions: correctness of the final answer, accuracy of problem comprehension, correctness of numerical calculations, and provision of a clear answer, weighted at 1:1:2:2, with the final score normalized to the range [-1, 0]. The prompt structure is consistently applied across all backbones and both prompts.

## G DETAILED FLOPS CALCULATION

This section provides a detailed breakdown of the Floating Point Operations (FLOPs) required for the Genius baseline and our proposed LATENTSEEK method, using LLaMA3.1-8B-Instruct as the backbone model.

Table 21: LATENTSEEK (Self) Hyperparameters on AIME2024. lr: learning rate. $\rho$: fraction ratio

| methods | model | max len | prompt idx | #GPU | lr | optimizer | $\rho$ | dtype | Max Step |
|---|---|---|---|---|---|---|---|---|---|
| LATENTSEEK (Self) | Qwen2-7B-Instruct | 1024 | 1 | 1 4090 | 0.03 | Adam | 0.1 | bf16 | 10 |
| LATENTSEEK (Self) | Qwen2-7B-Instruct | 1024 | 2 | 1 4090 | 0.03 | Adam | 0.1 | bf16 | 10 |
| LATENTSEEK (Self) | Qwen2.5-1.5B-Instruct | 1024 | 1 | 1 3090 | 0.03 | Adam | 0.1 | bf16 | 10 |
| LATENTSEEK (Self) | Qwen2.5-1.5B-Instruct | 1024 | 2 | 1 3090 | 10 | Adam | 0.3 | bf16 | 10 |
| LATENTSEEK (Self) | Qwen2.5-7B-Instruct | 1024 | 1 | 1 4090 | 0.05 | Adam | 0.2 | bf16 | 10 |
| LATENTSEEK (Self) | Qwen2.5-7B-Instruct | 1024 | 2 | 1 4090 | 0.05 | Adam | 0.2 | bf16 | 10 |
| LATENTSEEK (Self) | Qwen2.5-14B-Instruct | 1024 | 1 | 1 A100 | 0.03 | Adam | 0.1 | bf16 | 10 |
| LATENTSEEK (Self) | Qwen2.5-14B-Instruct | 1024 | 2 | 1 A100 | 0.03 | Adam | 0.1 | bf16 | 10 |
| LATENTSEEK (Self) | Llama-3.1-8B-Instruct | 1024 | 1 | 1 4090 | 0.03 | Adam | 0.1 | bf16 | 10 |
| LATENTSEEK (Self) | Llama-3.1-8B-Instruct | 1024 | 2 | 1 A100 | 0.03 | Adam | 0.2 | bf16 | 10 |

Table 22: LATENTSEEK (PSRM) Hyperparameters on AIME2024. lr: learning rate. $\rho$: fraction ratio

| methods | model | max len | prompt idx | #GPU | lr | optimizer | $\rho$ | dtype | Max Step |
|---|---|---|---|---|---|---|---|---|---|
| LATENTSEEK (PSRM) | Qwen2-7B-Instruct | 1024 | 1 | 1 3090 | 0.03 | Adam | 0.1 | bf16 | 10 |
| LATENTSEEK (PSRM) | Qwen2-7B-Instruct | 1024 | 2 | 1 3090 | 0.03 | Adam | 0.1 | bf16 | 10 |
| LATENTSEEK (PSRM) | Qwen2.5-1.5B-Instruct | 1024 | 1 | 1 A100 | 0.03 | Adam | 0.1 | bf16 | 10 |
| LATENTSEEK (PSRM) | Qwen2.5-1.5B-Instruct | 1024 | 2 | 1 A100 | 0.03 | Adam | 0.1 | bf16 | 10 |
| LATENTSEEK (PSRM) | Qwen2.5-7B-Instruct | 1024 | 1 | 1 3090 | 0.03 | Adam | 0.1 | bf16 | 10 |
| LATENTSEEK (PSRM) | Qwen2.5-7B-Instruct | 1024 | 2 | 1 4090 | 0.05 | Adam | 0.2 | bf16 | 10 |
| LATENTSEEK (PSRM) | Qwen2.5-14B-Instruct | 1024 | 1 | 1 A100 | 0.03 | Adam | 0.1 | bf16 | 10 |
| LATENTSEEK (PSRM) | Qwen2.5-14B-Instruct | 1024 | 2 | 1 A100 | 0.03 | Adam | 0.1 | bf16 | 10 |
| LATENTSEEK (PSRM) | Llama-3.1-8B-Instruct | 1024 | 1 | 1 4090 | 0.03 | Adam | 0.1 | bf16 | 10 |
| LATENTSEEK (PSRM) | Llama-3.1-8B-Instruct | 1024 | 2 | 1 A100 | 0.03 | Adam | 0.1 | bf16 | 10 |

Table 23: Prompt for answer correctness check

```
f"{math_prefix}"
"INSTRUCTIONS:"
"Your task is to determine whether the provided answer is
correct."
"Think through the verification process carefully and
logically."
"IMPORTANT RULES:"
"1. Do NOT analyze the steps or methods used to arrive at the
answer."
"2. Only evaluate the final answer's correctness."
"3. Your response must strictly follow the required format:"
f"- If the answer is correct, respond with:
'{VERA_ANSWER_SYMBOL}True'."
f"- If the answer is incorrect, respond with:
'{VERA_ANSWER_SYMBOL}False'."
```

Table 24: Prompt for calculation check

```
f"{math_prefix}"
"INSTRUCTIONS:"
"1. EXTRACT CALCULATION EXPRESSIONS: Extract all the
mathematical calculations from the PROPOSED SOLUTION."
"2. INDEPENDENT RECOMPUTATION: Break down the calculations
step-by-step and recompute them."
f"3. VERIFY: Compare your recomputation with the
PROPOSED SOLUTION. If any discrepancy is found, output
'{VERA_ANSWER_SYMBOL}False'. If all steps are correct, output
'{VERA_ANSWER_SYMBOL}True'."
"NOTE: You ONLY need to check calculations(like 1 + 1 = 2, 2 *
3 = 6, etc). Ignore standalone numbers(like 1, 2, 3, etc) that
are not part of a computation."
```

Table 25: Prompt for understanding check

```
f"{math_prefix}"
"INSTRUCTIONS:"
"1. PROBLEM INTERPRETATION:"
" - Assess if the proposed solution clearly understands the
problem statement."
" - Ensure that the proposed solution addresses all relevant
aspects of the problem, without ignoring any key detail."
" - Flag if the solution misinterprets or overlooks the
problem's core requirements or scope."
"2. ALIGNMENT WITH THE TASK:"
" - Verify that the solution responds to the specific question
or task outlined in the problem statement."
" - Ensure that the solution does not deviate from the
problem's context or provides an unrelated answer."
" - Check if any critical parts of the problem have been
misinterpreted or neglected."
"3. TERMINATION PROTOCOL:"
" - If the solution clearly misinterprets or fails to address
the problem correctly, stop and respond in the exact format:"
f" - '{VERA_ANSWER_SYMBOL}False'"
" - If the solution accurately captures the problem statement
and aligns with the required solution, respond in the exact
format:"
f" - '{VERA_ANSWER_SYMBOL}True'"
"EXAMPLES:"
"[Case 1] Problem: A shop is selling a drink at 1.5 times the
original price. If the original price is $10, what is the new
price?"
" Solution: The new price is 1.15 * $10 = $11.50."
" Assessment: The solution misinterprets the problem by
calculating 1.15 times the original price instead of 1.5
times."
f" Result: '{VERA_ANSWER_SYMBOL}False'"
"[Case 2] Problem: The second cup of coffee is half price. If
the first cup costs $5, how much is the second cup?"
" Solution: The second cup costs $5 * 0.5 = $2.50."
" Assessment: The solution correctly interprets the price as
half the original price for the second cup."
f" Result: '{VERA_ANSWER_SYMBOL}True'"
"[Case 3] Problem: A pizza has a radius of 8 inches. What is
the area of the pizza?"
" Solution: The area is $\pi r^2$, where r = 4 inches. The area is
$16\pi$ square inches."
" Assessment: The solution misinterprets the formula for the
area of a circle by using the radius incorrectly."
f" Result: '{VERA_ANSWER_SYMBOL}False'"
"[Case 4] Problem: A train is moving at 60 km/h towards the
east. What is its velocity after 2 hours?"
" Solution: The velocity is 120 km/h west."
" Assessment: The solution correctly calculates the speed, but
misinterprets the direction as west instead of east."
f" Result: '{VERA_ANSWER_SYMBOL}False'"
"CRITICAL REQUIREMENTS:"
"- Assess whether the solution addresses all parts of the
problem."
"- Ensure the solution does not deviate from the problem's
intent."
"- Use exact output formats specified, showing no tolerance for
misinterpretations."
```

Table 26: Prompt for answer completeness check

```
f"{math_prefix}"
"INSTRUCTIONS:"
"Your task is to verify whether the solution provides a
complete and final answer."
"Follow these rules carefully:"
"1. Check if the solution reaches a clear and definitive final
answer."
"2. The answer must not be left incomplete, such as:"
" - Ending with an unresolved expression or formula instead of
a computed result."
" - Missing a conclusion or final statement explicitly stating
the final answer."
"3. If the solution is incomplete or lacks a final answer,
immediately stop checking further and respond in the exact
format:"
f" - '{VERA_ANSWER_SYMBOL}False'"
"4. If the solution is complete and provides a final, explicit
answer, respond in the exact format:"
f" - '{VERA_ANSWER_SYMBOL}True'"
"Examples:"
"Example 1:"
"final answer: 8."
f"Your response: '{VERA_ANSWER_SYMBOL}True' (The solution
provides a final, definitive answer of 8.)"
"Example 2:"
"final answer: The area of the circle is πr², where r = 4."
f"Your response: '{VERA_ANSWER_SYMBOL}False' (The answer ends
with an unresolved formula, not a computed result.)"
"Example 3:"
"final answer: This question does not have an answer or I
cannot solve this problem."
f"Your response: '{VERA_ANSWER_SYMBOL}False' (The solution lacks
a clear, final answer.)"
```

Table 27: Estimated FLOPs for a single forward pass of the LLaMA3.1-8B backbone model with a context length of 512 tokens. The total is derived by summing the operational costs for both the prefill and decode stages.

| Prefill Stage (512 tokens) | | Decode Stage (512 tokens) | |
|---|---|---|---|
| **Component** | **FLOPs** | **Component** | **FLOPs** |
| q_proj | 17.2G | q_proj | 33.6M |
| k_proj | 4.3G | k_proj | 8.4M |
| v_proj | 4.3G | v_proj | 8.4M |
| out_proj | 17.2G | out_proj | 33.6M |
| gate_proj | 60.1G | gate_proj | 117M |
| up_proj | 60.1G | up_proj | 117M |
| down_proj | 60.1G | down_proj | 117M |
| qk_matmul | 2.1G | qk_matmul | 4.2M |
| sv_matmul | 2.1G | sv_matmul | 4.2M |
| softmax | 41.9M | softmax | 81.9K |
| attn_norm | 14.7M | attn_norm | 28.7K |
| mlp_norm | 14.7M | mlp_norm | 28.7K |
| attn_add | 2.1M | attn_add | 4.1K |
| mlp_add | 2.1M | mlp_add | 4.1K |
| mlp_act | 4.2M | mlp_act | 8.2K |
| lm_head | 525M | lm_head | 525M |

## G.1 TOTAL FLOPs FOR GENIUS

On the GSM8K dataset, the Genius framework involoves a full training phase on 100000 instances followed by testing on 1319 instances. We assume a backward pass is approximately twice as computationally expensive as a forward pass.

- **Training FLOPs:** For each of the 100000 instances, one forward pass and one full backward pass are required.

$$\text{FLOPs}_{\text{train}} = 100000 \times (\text{FLOPs}_{\text{forward}} + 2 \times \text{FLOPs}_{\text{forward}}) = 300000 \times \text{FLOPs}_{\text{forward}}$$

- **Testing FLOPs:** For each of the 1319 instances, one forward pass is required.

$$\text{FLOPs}_{\text{test}} = 1319 \times \text{FLOPs}_{\text{forward}}$$

$$\begin{aligned}
\text{FLOPs}_{\text{Genius}} &= \text{FLOPs}_{\text{train}} + \text{FLOPs}_{\text{test}} \\
&= (300000 + 1319) \times (2.29 \times 10^{11}) \\
&\approx 6.90 \times 10^{16} \text{FLOPs}
\end{aligned}$$

## G.2 TOTAL FLOPs FOR LATENTSEEK

LATENTSEEK operates exclusively at test time on 1319 instances. Each iteration involves two forward passes (one for generation, one for reward evaluation) and one partial backward pass through only the LM head. From Table 9, the average number of iterations on GSM8K is 1.27.

- **Number of Forward Passes:** 1319 instances $\times$ 1.27 iter/inst $\times$ 2 forward/iter $\approx 3350$
- **Number of Backward Passes:** 1319 instances $\times$ 1.27 iter/inst $\times$ 1 forward/iter $\approx 1675$
- **Cost of Backward Pass:** The backward pass is only through the LM head (525 MFLOPs forward).

$$\text{FLOPs}_{\text{backward\_LATENTSEEK}} = 2 \times (525 \times 10^6) = 1.05 \times 10^9 \text{ FLOPs}$$

The total extimated FLOPs for LATENTSEEK is:

$$\begin{aligned}
\text{FLOPs}_{\text{LATENTSEEK}} &= (3350 \times \text{FLOPs}_{\text{forward}}) + (1675 \times \text{FLOPs}_{\text{backward\_LATENTSEEK}} \\
&= (3350 \times 2.29 \times 10^{11}) + (1675 \times 1.05 \times 10^9) \\
&\approx 7.67 \times 10^{14} + 1.76 \times 10^{12} \\
&\approx 7.69 \times 10^{14} \text{FLOPs}
\end{aligned}$$

## G.3 EFFICIENCY THRESHOLD ANALYSIS

We calculate the number of inference instances ($x$) at which the total computational cost of using the pre-trained Genius framework would equal the cost of using LATENTSEEK. This threshold is found by solving the following equation, where the left side represents the total cost of Genius and the right side represents the total cost of LATENTSEEK for $x$ inferences:

$$6.90 \times 10^{16} + x \cdot (2.29 \times 10^{11}) = 1.27 \times (2 \times 2.29 \times 10^{11} + 1.05 \times 10^9) \cdot x$$

Solving for $x$ yields the threshold: $x \approx 1.94 \times 10^5$. This calculation demonstrates that LATENTSEEK remains more efficient than the Genius method up to approximately $1.94 \times 10^5$ inference instances, which exceeds the size of Genius's training set ($1.00 \times 10^5$ examples).

## G.4 COMPARISON WITH ICOT

To the best of our knowledge, all latent reasoning methods experience a training stage, and iCoT (Deng et al., 2024) is one of the representative. We perform FLOPs comparison with iCoT using Llama3.1-8B-Instruct evaluated on the GSM8K test set. As shown in Table 28, the **total FLOPs of LatentSeek are roughly $10^4$ times smaller than those of iCoT**. This efficiency comes from **eliminating the need of training**: LatentSeek performs latent updates *only at test time*, and each update requires just *a single forward–backward pass of the language-model head*, making the procedure efficient. Moreover, as shown in Table 9, LatentSeek typically converges in fewer than two update iterations during inference.

Table 28: FLOPs comparison with iCoT (Deng et al., 2024)

| Name | Type | # Train Dataset | # Train Epoch | # Test Dataset | Acc (%) | Total FLOPs |
|------|------|-----------------|---------------|----------------|---------|-------------|
| LatentSeek | Latent | 0 | 0 | 1.3K | 83.7 | $7.69 \times 10^{14}$ |
| iCoT | Latent | 37.8K | 200 | 1.3K | 47.08 | $5.19 \times 10^{18}$ |

# H  LATENT SPACE TRAJECTORIES

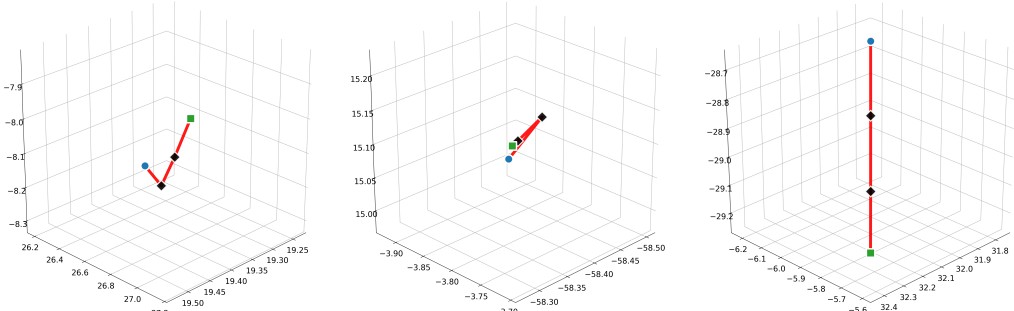

Figure 8: Examples of latent-space optimization trajectories. Blue: initial latent vector; Black: intermediate steps; Green: final latent vector; Red: optimization trajectory. For more trajectories, please refer to Figure 9 and Figure 10

To better understand the optimization dynamics within the continuous latent space, we conduct a visualization study of latent-vector trajectories. Specifically, for each token position $t$ and optimization step $k$, we collect the corresponding latent representations $z_t^{(k)}$ and apply Principal Component Analysis (PCA) to project these high-dimensional vectors into a three-dimensional space.

**As shown in Figure 9 and Figure 10 below**, we visualize the projected trajectories of token latents across different positions (1st to 80th). Furthermore, Figure 8 highlights three representative patterns of latent-space optimization. The Blue dot denotes the initial latent vector, the Black dots indicate intermediate latent states, the Green dot marks the final latent vector, and the Red line traces the optimization trajectory. As illustrated in the figures, some token-specific latents exhibit zig-zag search, others follow a consistent unidirectional trajectory, and still others trace an V-shaped search path.

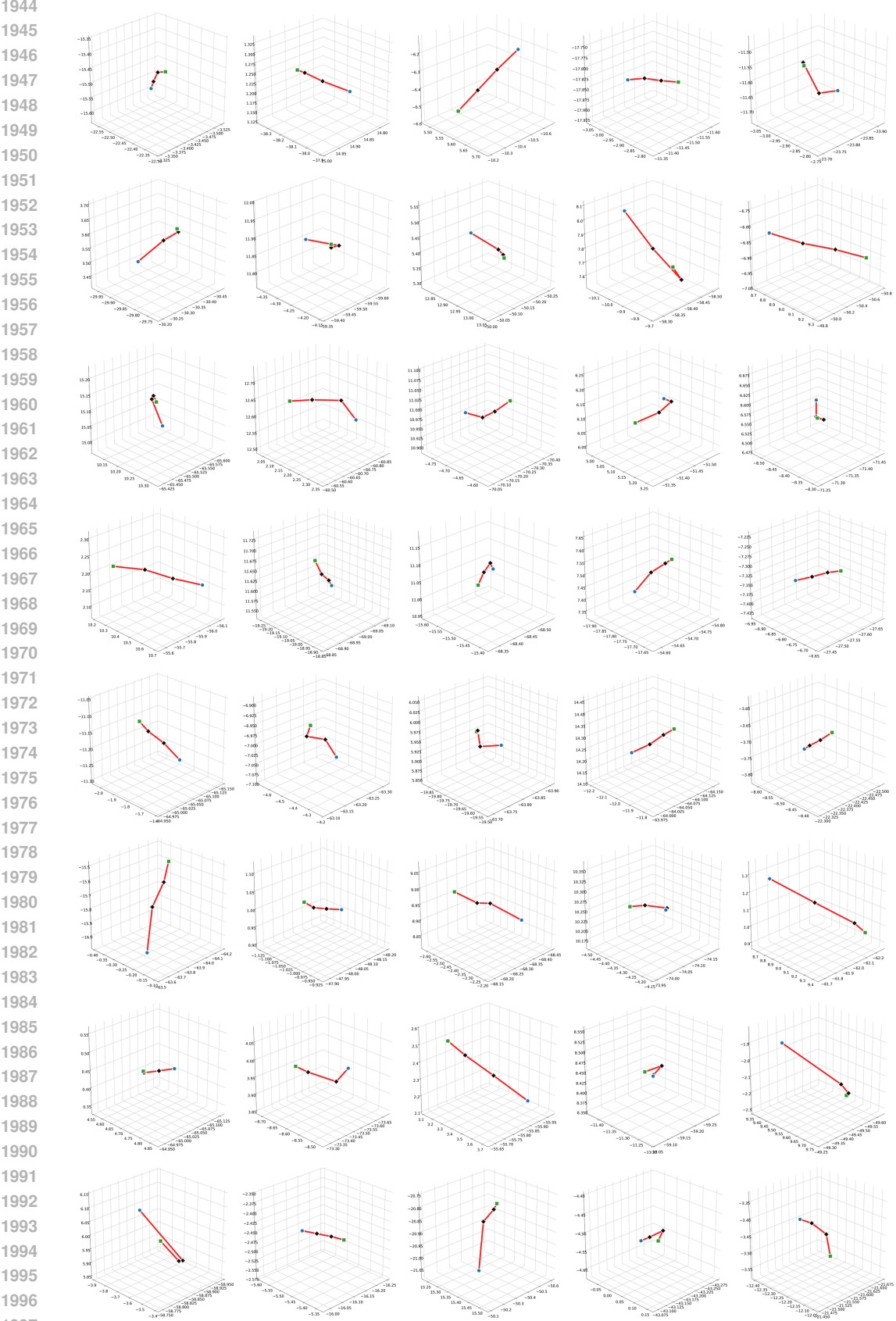

Figure 9: Latent-space trajectories for tokens at positions 1–40. Blue: initial latent vector; Black: intermediate steps; Green: final latent vector; Red: trajectory.

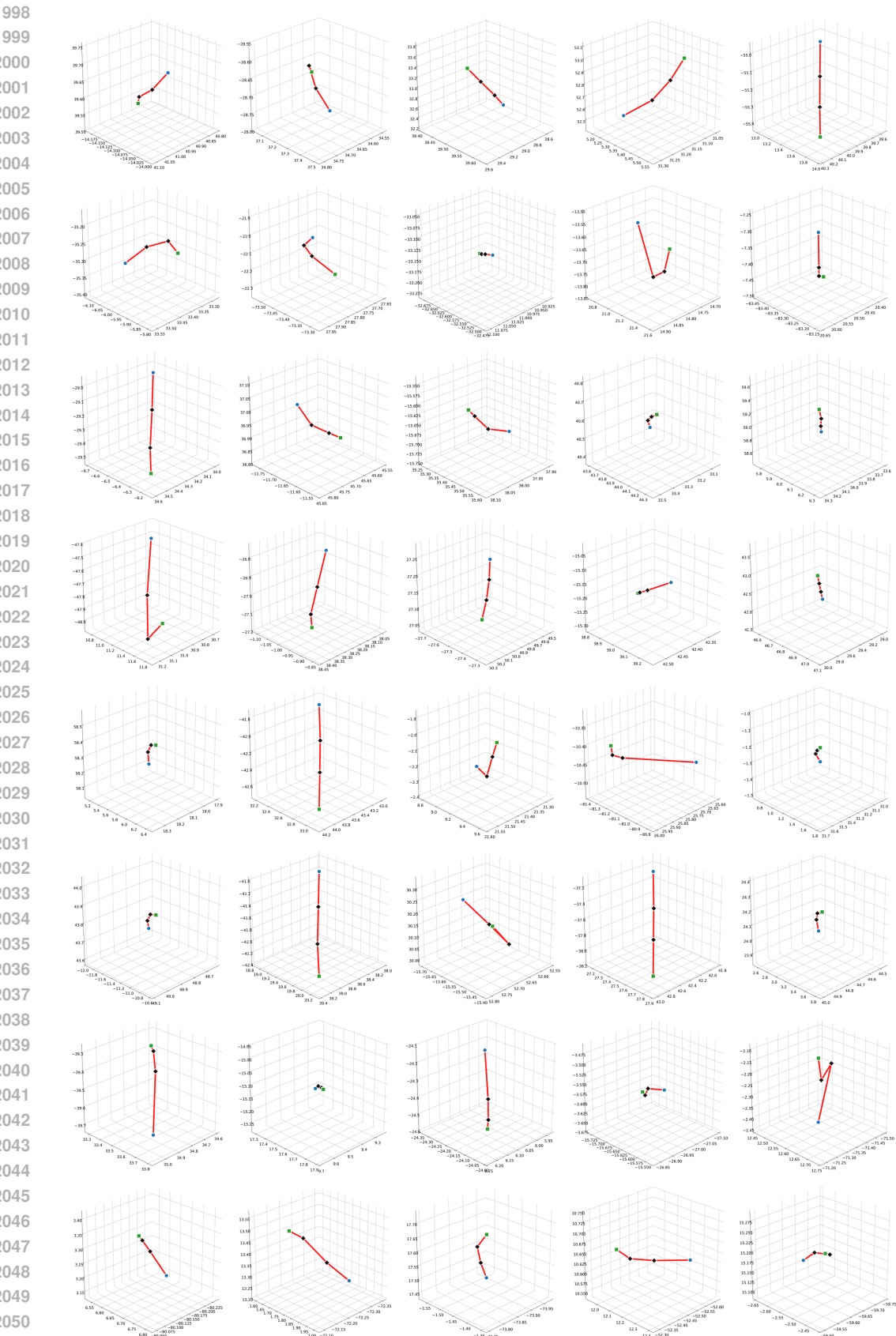

Figure 10: Latent-space trajectories for tokens at positions 41–80. Blue: initial latent vector; Black: intermediate steps; Green: final latent vector; Red: trajectory.

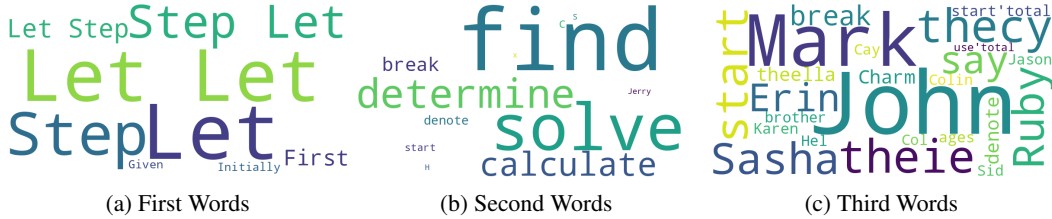

|                  |                   |                  |
| :--------------: | :---------------: | :--------------: |
| (a) First Words  | (b) Second Words  | (c) Third Words  |

Figure 11: Wordclouds of the first three words of the generated sequence using prompt 2 on the GSM8k dataset with Llama3.1-8B-Instruct as the backbone.

## I  QUALITATIVE ANALYSIS AND CASE STUDIES

To gain a deeper understanding of model-generated responses, we analyze the distribution of the first three tokens produced under Prompt 1 in the GSM8K dataset Cobbe et al. (2021), using the Llama3.1-8B-Instruct model. These distributions are visualized in the word cloud shown in Figure 11. The appearance of nonsensical or obscure tokens—such as "thecy," "theella," and unrelated proper names like "Mark" and "John"—suggests that the reasoning processes employed by language models can diverge significantly from human reasoning, despite generating superficially plausible outputs. Notably, consistent patterns emerge: the first generated tokens are often prepositions (e.g., "let"), the second tokens are primarily verbs (e.g., "find," "solve"), and the third tokens are predominantly proper names, These findings suggest that **optimal reasoning paths for language models may differ from human strategies and could be more effectively modeled in latent space.**

We present several representative cases in Tables 29 to 35 to demonstrate the divergence between machine reasoning patterns and human cognition processes. These cases reveal that while our method (*OURS*) generates linguistically anomalous expressions such as "total downloads of downloads", "let need calculate to calculate" and "let'll more understand it", it still achieves correct solutions. This suggests that even if the decoded tokens from our updated embeddings lack syntactic and semantic coherence, the logic and answer of the final generated output remain accurate. Notably, although the baseline model produces erroneous answers that appear fundamentally distinct from ours (*OURS*) in textual form, their latent representations exhibit structural proximity - through minimal adjustments to embeddings in the latent space, the original output (*ORIGINAL MODEL*) can be transfromed into correct solutions (*OURS*). These empirical evidences yield two critical implications. First, it suggests that **the reasoning paths employed by machines are fundamentally distinct from those of humans**. Second, **while model errors may appear substantially deviant from correct answers in textual outputs, they may maintain significant proximity to valid solutions in latent space.**

Table 29: Case 1

```
QUESTION: A new program had 60 downloads in the first month.
The number of downloads in the second month was three times as
many as the downloads in the first month, but then reduced by
30% in the third month. How many downloads did the program have
total over the three months?
TRUE ANSWER: 366
ORIGINAL MODEL: {"thought process": "To find the total number
[...] Finally, we add up the number of downloads in all three
months to get the total number of downloads.", "final answer":
213}
OURS: {"thought process": "First solve the total downloads of
downloads, the three months, we need calculate the downloads
of the second month and the third month. [...] Now, we can
calculate the total downloads over the three months, total
downloads = 60 + 180 + 126 = 366.", "final answer": 366}
```

Table 30: Case 2

QUESTION: Jen enters a lottery by picking 4 distinct numbers
from $S = \{1,2,3,\cdots,9,10\}$. 4 numbers are randomly chosen from
$S$. She wins a prize if at least two of her numbers match the
chosen numbers, and wins the grand prize if all four match.
Find the conditional probability of winning the grand prize
given that she won any prize.
TRUE ANSWER: 116
ORIGINAL MODEL: {"thought process": "To solve this problem
[...] Finally, we can add m and n to get the final answer.",
"final answer": 5}
OURS: {"thought process": "First find this problem, let need
calculate to calculate the probability number of possible to
can choose a prize and the number of possible to can choose the
grand prize. [...] Therefore, the answer is $1 + 115 = 116$.",
"final answer": 116}

Table 31: Case 3

QUESTION: Two trains leave San Rafael at the same time. They
begin traveling westward, both traveling for 80 miles. The next
day, they travel northwards, covering 150 miles. What's the
distance covered by each train in the two days?
TRUE ANSWER: 230
ORIGINAL MODEL: To solve this problem, we need to break it down
into steps. [...] So, the distance covered by each train in the
two days is 197.5 miles. \boxed{197.5}
OURS: Let find this,, let'll more understand it down step
two andLet 1: BothThe the trains are same route west both
first time on80ward) on same and and can consider they the
travel travel same distance of So's denote the common xd' they
both 80 miles on x have write up an following: [...] Since
both trains travel the same distance in each direction, the
distance covered by each train is 230 miles. The final answer
is: \boxed{230}

Table 32: Case 4

QUESTION: Carlos is planting a lemon tree. The tree will cost
$90 to plant. Each year it will grow 7 lemons, which he can
sell for $1.5 each. It costs $3 a year to water and feed the
tree. How many years will it take before he starts earning
money on the lemon tree?
TRUE ANSWER: 13
ORIGINAL MODEL: {"thought process": "To find out how many years
[...] We will continue this process for each subsequent year,
adding the maintenance cost and calculating the income until
the total income is greater than the total cost.", "final
answer": 7}
OURS: {"thought process": "Let determine the when long years
Carlos will take before the to earn earning money from his
tree tree, I first to find the revenue revenue for planting the
maintaining the tree. [...] Therefore, it will take 13 years
for Carlos to start earning money on the lemon tree.", "final
answer": 13}

Table 33: Case 5

QUESTION: Gunter is trying to count the jelly beans in a jar. He asks his friends how many they think are in the jar. One says 80. Another says 20 more than half the first one. A third says 25% more than the first one. What is their average guess?
TRUE ANSWER: 80
ORIGINAL MODEL: {"thought process": "To find the average guess [...] After finding the individual guesses, we can add them up and divide by the number of guesses to find the average.", "final answer": 76}
OURS: {"thought process": "First find the average of, we first to calculate calculate each number guesses of The first friend guesses 80. [...] Then we add the three guesses together, 80 + 60 + 100 = 240. Finally, we divide by 3 to get the average, 240 / 3 = 80.", "final answer": 80}

Table 34: Case 6

QUESTION: Frankie watches TV after he finishes his homework every night. On Monday and Tuesday, he watched a 1-hour episode of his favorite show each night. On Wednesday, he watched a few episodes of a 30-minute show. On Thursday, he finished homework early and watched a 1-hour episode and a 30-minute show. On Friday, he got to stay up late for the weekend, so he watched two 1-hour episodes. If he watched 7 hours of TV in all, how many 30-minute episodes did he watch on Wednesday?
TRUE ANSWER: 3
ORIGINAL MODEL: To determine how many 30-minute episodes Frankie watched on Wednesday, [...] So, Frankie watched \boxed{197.5} 30-minute episodes on Wednesday.
OURS: Let determine how many 30-minute episodes Frankie watched on Wednesday, let will to break the total amount he spent watching TV from night and then solve the the much time was spent on Wednesday specifically1, we's summarize the time time Frankie watching TV each day: [...] Thus, the number of 30-minute episodes Frankie watched on Wednesday is \boxed{3}.

Table 35: Case 7

QUESTION: Lloyd has an egg farm. His chickens produce 252 eggs per day and he sells them for $2 per dozen. How much does Lloyd make on eggs per week?
TRUE ANSWER: 294
ORIGINAL MODEL: {"thought process": "To find out how much Lloyd makes on eggs per week, [...] Finally, we multiply the number of dozens by the price per dozen to get the total amount Lloyd makes in a week.", "final answer": 5880}
OURS: {"thought process": "First calculate Lloyd how much Lloyd makes per eggs per week, first need calculate to determine out the many eggs he produces in a week. [...] Total amount per week = 147 * 2 = $294. Therefore, Lloyd makes $294 per week on eggs..", "final answer": 294}

## J    COMPUTATIONAL RESOURCES

All experiments are performed with a single A100, or a single L40, or a single 4090, or a single 3090.

## K    THE USE OF LARGE LANGUAGE MODELS (LLMS)

Our use of Large Language Models (LLMs) was strictly limited to polishing the language and generating figures for the manuscript. All underlying research and intellectual content of this paper, including the LATENTSEEK framework, its theoretical foundations, experimental design, and the analysis of results, was completed entirely by the authors without assistance from LLMs.

