# OpenReview forum: "Reasoning via Test-Time Instance-Level Policy Gradient in Latent Space"
_ICLR.cc/2026/Conference — ICLR 2026 Conference Desk Rejected Submission_

### Official Review · Reviewer_EsPa · 2025-10-23

**Soundness:** 3
**Presentation:** 3
**Contribution:** 3
**Rating:** 4
**Confidence:** 4

**Summary:**

This paper proposes LATENTSEEK, a framework that enhances LLM reasoning by performing test-time policy gradient optimization in the latent space rather than over explicit tokens. Instead of fine-tuning model parameters or relying on handcrafted prompts, LATENTSEEK performs lightweight, instance-level adaptation: it iteratively updates latent representations (pre-decoding states) guided by a self-generated reward.

**Strengths:**

1. The idea of latent-space optimization without parameter updates is noval and bridges reinforcement-based adaptation with test-time inference.

2. Improving the quality of CoT paths by optimizing the latent space seems interesting, especially since the authors mention that the number of iterations is usually less than 3. This provides insight into test-time scaling.

3. The experimental results are good.

**Weaknesses:**

1. While the authors highlight that latent updates yield incoherent tokens, they do not analyze what these updates actually do in embedding space (e.g., directionality, similarity drift, or semantic shifts).

2. The self-generated reward signal (Eq. 8) is vaguely defined and may not correlate with correctness. No ablation isolates its contribution or evaluates instability across tasks.

3. Although the paper references “Appendices C–D for theoretical justification,” the main text does not clarify why the independence assumption between latent representations (Eq. 7) holds or how gradient estimation avoids collapse.

4. Since the reward takes only binary values (0/1), it lacks continuous gradient information and therefore cannot provide fine-grained directional guidance.
As a result, the optimization may rely more on stochastic noise than on the objective reflected by the reward itself. Moreover, if the model’s self-evaluation were sufficiently accurate, one could simply adopt that evaluation as the final decision; if it is inaccurate, then its validity as an optimization signal becomes questionable. This reveals an inherent inconsistency: the model’s ability to assess its output after generation does not necessarily translate into an effective signal for guiding latent optimization before generation.

5. The model optimizes only the first ρT (e.g., 20%) of the latent variables, with the number of iterations limited to ≤3. Such mild perturbation may be functionally equivalent to a single stochastic exploration step, where improvements arise from increased generative diversity rather than genuine gradient guidance. It is recommended to include control experiments with random perturbation (Random Perturbation / Reverse Reward) to rule out this factor.

6. The proposed method first initializes the reasoning process with an explicit CoT trajectory and then performs optimization in the latent space. This design appears to rely heavily on the quality of the initial CoT trajectory. If the initial reasoning path deviates from the correct direction, performing only about three iterations of updates on merely 20% of the latent representations may be insufficient to substantially correct the reasoning trajectory.

7. In most reasoning sequences, errors tend to occur in the middle or later stages rather than at the beginning. Therefore, optimizing only the early latent tokens may not effectively address the actual sources of error. Overall, the method seems to perform slight adjustments or perturbations to the initial reasoning direction. I remain skeptical that its effectiveness arises mainly from random perturbations to the initial direction, rather than from genuine reinforcement learning with stable and informative reward signals—especially given that the reward employed in the paper appears highly noisy.

**Questions:**

I prefer the style and insight of the paper, but there are some issues that lead me to have some reservations. I lean more towards borderline (but there is no option for a score of 5). If the authors can address some of my concerns, I would be willing to raise my rating to acceptance.

---

> ### Author Response · Authors · 2025-11-20
> **Response to Reviewer EsPa**
>
> We thank the reviewer for the insightful and detailed review!
>
>
> > W1. While the authors highlight that latent updates yield incoherent tokens, they do not analyze what these updates actually do in embedding space (e.g., directionality, similarity drift, or semantic shifts).
>
>
> Thanks for pointing out! We'd like to clarify as follows:
>
>
>
> - **Latent Space Trajectories (See newly added Figure 8, 9, and 10 in Appendix H):** We conduct a visualization study of latent-vector trajectories. **As shown in Appendix H, we provide the corresponding trajectory plots.** To obtain these trajectories, we first reduce the latent vectors to three dimensions using PCA, and then visualize the resulting paths for each token individually. These trajectories illustrate how directions evolve in latent space: some latent vectors move far from the origin in unidirection, while others follow zig-zag paths, etc.
> - **Moving to High-Reward Region (See newly added Figure 5 in Appendix E.6):** From a macro perspective, **the direction of the move in latent space is the direction towards a high-reward region**. As shown in the newly added Figure 5 of the manuscript, **the mean reward consistently increases across all models** as the number of optimization steps increases.
> - **Low-level view of Directions:** Specifically, the policy gradient (Eq. 7) determines the exact direction of the update. Noticed that, our self-reward mechanism is design to generate a reward in $[-1, 0]$ (see Sec 3.1 of the revised manuscript). Thus, when a decoded trajectory $x$ yields a negative reward $R(x,c)$, the update $\nabla_z J(z)$ actively **pushes the latent vector $z$ away from the last latent state that decodes to the previous low-reward token $x$**. This effectively lowers the probability of the previously selected tokens. Furthermore, which token $x'$ is chosen as the replacement **is not random**; it is determined by moving along the gradient direction on the **pre-trained  model's latent space**, and thus the gradient direction indeed matters.
> - **Drift from Semantic Perspective:** We analyzed this exact behavior qualitatively in **Appendix I and Figure 11**, discovering a slight sign of semantic shift as mentioned by the reviewer. We found that when optimizing the initial tokens, the first token is biased to become a **preposition** (e.g., **"Let"**) and the second token is biased to become a **verb** (e.g., **"find", "solve"**). However, we haven't found a clear pattern for subsequent tokens yet.
>
> ---

---

> > ### Author Response · Authors · 2025-11-20
> > **Response to Reviewer EsPa (2)**
> >
> > > W2. The self-generated reward signal (Eq. 8) is vaguely defined and may not correlate with correctness. No ablation isolates its contribution or evaluates instability across tasks.
> >
> >
> >
> > We apologize for the vagueness and thank the reviewer for this insightful comment!
> >
> > 1. **Reward Methanism Definition**:  The self reward is described in **Sec 3.1**.  To generate the reward, the pre-trained model is **prompted** to evaluate the self-generated answer according to multiple criterias, with **a score between -1 and 0** as final output (See Appendix F or Sec 3.1). The self-reward mechanism uses **the same prompt across all tasks** while still delivering consistent performance improvements, indicating strong **stability on tasks**. *We have updated Sec 3.1 for clearer description*.
> > 2. **Additional Ablation Study**: We perform an **ablation study on the GSM8K test set** using prompt 2 (the JSON prompt, defined in the paper), demonstrating the importance of the self-generated reward signal:
> >
> >     **Explanation of Settings:**
> >     - **LatentSeek (Self)**: Using self reward mechanism.
> >     - **LatentSeek (Constant Reward)**: This baseline **ablates the reward signal entirely**. In this setting, the optimization is run for the full number of iterations using **a constant reward of –1 at every step**. This isolates *the effect of an unguided search from that of our reward-guided approach*. Note that the reason to choose a negative constant is that when the reward is non-negative, the update simply pushes the latent state toward regions that increase the likelihood of the *current* tokens. Under the argmax decoding used in our framework, this causes the reasoning trajectory to remain unchanged.
> >
> >
> >
> >     **Ablation Results:**
> >
> > | Backbone | Qwen2-7B-Instruct  | Qwen2.5-1.5B-Instruct|Qwen2.5-7B-Instruct | Qwen2.5-14B-Instruct |LLaMA3.1-8B-Instruct | Average |
> > |----------|-------|-----------|------|-----|-----|---|
> > |   LatentSeek (Constant)    |   60.19  |   29.49       | 64.52 |  90.67   |69.98|62.97 |
> > |    LatentSeek (Self)     |  $80.21_{+20.02}$   |  $44.20_{+14.71}$  |  $85.06_{+20.54}$    |$92.72_{+2.05}$ | $83.70_{+13.72}$ | $77.18_{+14.21}$ |
> >
> > As shown above, when we remove the guidance of the reward model, using a constant reward *results in an average performance drop of 14.21*. This drop indicates that the the self-reward guidance is *essential and useful*.
> >
> > **We've added this experiment to Sec 3.6 in our revised manuscript.**
> >
> >
> > 3. **Reward Model Correctness**: Recent research has demonstrated that large language models can serve as reliable reward sources [1, 2]. To further address the reviewer's concern about whether the reward correlates with correctness, we conduct an additional analysis on the GSM8K test set using the prompt 2 (the JSON prompt, defined in the paper). Specifically, we collect **all explored sequences**, including intermediate reasoning steps as well as final generated answers, and evaluate the alignment between the reward and ground-truth correctness. We binarized our dense [-1, 0] reward to "Positive" (Reward > -0.2) and "Negative" (Reward <= -0.2) and compared this against the ground-truth correctness. This yields the following results:
> >
> >
> > |Model |Qwen2-7B-Instruct|Qwen2.5-1.5B-Instruct | Qwen2.5-7B-Instruct | Qwen2.5-14B-Instruct | Llama3.1-8B-Instruct|
> > | :---: | :---: | :---: | :---: | :---: |:---:|
> > | Reward Acc | 85.78\% |77.15\% | 80.17\%|89.83\% | 80.46\%|
> >
> > As shown above, in most cases, the *self reward mechanism is reliable*. In practice, the ability to judge an answer **cooperate with** the ability to generate one, and together they lead to substantially improved overall performance.
> >
> > **We've added this discussion to Sec 3.6 in our revised manuscript.**
> >
> >
> > [1] Yuan et al. 2024. Self-Rewarding Language Models.
> >
> > [2] Lifshitz et al. 2025. Multi-agent verification: Scaling test-time compute with multiple verifiers.
> >
> > ---

---

> ### Author Response · Authors · 2025-11-20
> **Response to Reviewer EsPa (3)**
>
> > W3. Although the paper references “Appendices C–D for theoretical justification,” the main text does not clarify why the independence assumption between latent representations (Eq. 7) holds or how gradient estimation avoids collapse.
>
>
>
> Thanks for pointing this out, this is important! We’ve added more explanation to **Sec 2.2 in the main text** to explain the independence assumption, including both theoretical motivation and empirical considerations:
>
> `The independence assumption is adopted for two main reasons. (1) Enlarged Exploration Space. Without this assumption, the autoregressive structure would force the optimization to concentrate on the first latent token, since all subsequent tokens are conditioned on it. This collapses the effective search space. By contrast, imposing independence decouples the latent components, yielding a substantially larger exploration space and a more flexible “launch pad” for generating the subsequent reasoning trajectory,` $\prod_{t=N+1}^{T} \pi(x_t \mid x_{<t}, \mathbf{c})$. `(2) Theoretical Guidance. Our analysis in Appendix C draws an analogy to Multi-Prover Interactive Proofs (MIP), and shows that the potential loss introduced by the independence assumption can be mitigated by both the base model's ability to faithfully generate the follow-up reasoning trajectory and the reward model’s ability to evaluate it.`
>
> ---

---

> > ### Author Response · Authors · 2025-11-20
> > **Response to Reviewer EsPa (4)**
> >
> > > W4. Since the reward takes only binary values (0/1), it lacks continuous gradient information and therefore cannot provide fine-grained directional guidance. As a result, the optimization may rely more on stochastic noise than on the objective reflected by the reward itself. Moreover, if the model’s self-evaluation were sufficiently accurate, one could simply adopt that evaluation as the final decision; if it is inaccurate, then its validity as an optimization signal becomes questionable. This reveals an inherent inconsistency: the model’s ability to assess its output after generation does not necessarily translate into an effective signal for guiding latent optimization before generation.
> >
> >
> > Thanks for pointing out! We'd like to clarify as follows:
> >
> >
> > 1. **The Self-Reward is Dense, Not Binary.** We believe there may be a slight misunderstanding. The self-reward mechanism is dense, providing a score between -1 and 0.
> > 2. **Comparison with Stochastic Exploration:** We provide a thorough comparison in the response to weakness 5 below. Please refer to weakness 5 below.
> > 3.
> >     > if the model’s self-evaluation were sufficiently accurate, one could simply adopt that evaluation as the final decision;
> >
> >      There is a small misunderstanding here, so we would like to clarify the role of the self-reward function. The self-reward function, $R(x, c)$, serves purely as an evaluator: it takes an answer $x$ and *returns a score* indicating the degree of correctness. Even if this evaluator has the capability to verify the responses, it can't *produce the correct answers directly*, and the model must still engage in a search process (like LatentSeek) to *find an $x$ that scores highly on $R(x, c)$*.
> >
> > 4. **Reward Accuracy**: As mentioned in our response to weakness 2, we conduct an ablation study of the reward mechanism and a quantitative analysis of our self-reward model's accuracy (the table is copied below for convenience of reading). This analysis demonstrate that our self-reward signal is **accurate in the majority of cases** and provides a **reliable signal for guiding the latent space optimization**, even if it is not perfect.
> >
> > |Model |Qwen2-7B-Instruct|Qwen2.5-1.5B-Instruct | Qwen2.5-7B-Instruct | Qwen2.5-14B-Instruct | Llama3.1-8B-Instruct|
> > | :---: | :---: | :---: | :---: | :---: |:---:|
> > | Reward Acc | 85.78\% |77.15\% | 80.17\%|89.83\% | 80.46\%|
> >
> >
> > 5. **Trends of the Reward Signal:** To further illustrate the effect of the self-reward mechanism, we plot the trajectory of the average reward over optimization steps. As shown in the newly updated **Figure 5 (Appendix E.6)** of the manuscript, **the mean reward consistently increases across all models as the number of optimization steps increases**. It demonstrates that our reward signal **can provide directional guidance** to the optimization process.
> >
> >
> > We've added the discussion of our self-reward mechanism to **Sec 3.6** and **Appendix E.6**.
> >
> > ---

---

> > > ### Author Response · Authors · 2025-11-20
> > > **Response to Reviewer EsPa (5)**
> > >
> > > > W5. The model optimizes only the first ρT (e.g., 20%) of the latent variables, with the number of iterations limited to ≤3. Such mild perturbation may be functionally equivalent to a single stochastic exploration step, where improvements arise from increased generative diversity rather than genuine gradient guidance. It is recommended to include control experiments with random perturbation (Random Perturbation / Reverse Reward) to rule out this factor.
> > >
> > > Thanks for pointing this out! We would like to clarify that although the method typically converges in fewer than three iterations (Table 9), the **gradient information remains essential**. To verify this, we conducted an additional experiment, comparing LatentSeek with Stochastic Exploration, as suggested by the reviewer:
> > >
> > >
> > > **Experimental Settings**
> > >
> > >
> > > * **Stochastic Exploration:** This baseline performs *single random perturbation updates*. Specifically, we use a **Brownian motion** with $\sigma^2=0.5, 0.75, 1.00$. This setting isolates the behavior of a purely unguided random walk.
> > >
> > >
> > > **Experimental Results:**
> > >
> > > |                                       | Qwen2 | Qwen2.5 | Qwen2.5 | Qwen2.5 | LLaMA3.1 | Average |
> > > | ------------------------------------- | ----- | ------- | ------- | ------- | -------- | ------- |
> > > |                                       | **7B**    | **1.5B**    | **7B**      | **14B**     | **8B**       | **-**       |
> > > | **GSM8K Prompt1**                  |       |         |         |         |          |         |
> > > | Stochastic Exploration ($\sigma^2=0.50$) | 66.19 | 64.22   | 88.60   | 92.33   | 49.73    | 72.21   |
> > > | Stochastic Exploration ($\sigma^2=0.75$) | 65.96 |  63.53  |  88.60 | 92.03 |    49.73 | 71.97  |
> > > | Stochastic Exploration ($\sigma^2=1.00$) | 65.13 | 63.76   |  89.08 | 91.88 |  50.57   |  72.08 |
> > > | LatentSeek                            | **84.38** | **70.89**   | **90.14**   | **92.49**   | **78.54**    | **83.29**   |
> > > | **GSM8K Prompt2**                    |       |         |         |         |          |         |
> > > | Stochastic Exploration ($\sigma^2=0.50$)| 64.32 | 16.68   | 65.60   | 92.04   | 67.85    | 61.30   |
> > > | Stochastic Exploration ($\sigma^2=0.75$) | 64.77 |  15.69  |  64.92 | 91.89 |   67.02  |  60.86 |
> > > | Stochastic Exploration ($\sigma^2=1.00$) | 63.84 |  16.07  | 66.34  | 92.12 |  66.72   | 61.02  |
> > > | LatentSeek                            | **80.21** | **44.20**   | **85.06**   | **92.72**   | **83.70**    | **77.18**   |
> > >
> > >
> > > As shown in the table, replacing our policy gradient with an unguided random walk leads to an average performance drop of **13.66 points**. This demonstrates that the improvements are not from mere generative diversity, underscoring the importance of gradient-based guidance.
> > >
> > > **We've added this experiment to Sec 3.5 in main text**.
> > >
> > > ---

---

> > > > ### Author Response · Authors · 2025-11-20
> > > > **Response to Reviewer EsPa (6)**
> > > >
> > > > > w6. The proposed method first initializes the reasoning process with an explicit CoT trajectory and then performs optimization in the latent space. This design appears to rely heavily on the quality of the initial CoT trajectory. If the initial reasoning path deviates from the correct direction, performing only about three iterations of updates on merely 20% of the latent representations may be insufficient to substantially correct the reasoning trajectory.
> > > >
> > > >
> > > >
> > > > Thanks for raising this insightful question! We'd like to respond to this question as follows:
> > > >
> > > > 1. **Initialization is crucial:** The reviewer is correct that initialization is important. It's true that in many optimization problems, a good initialization is crucial, such as the use of Kaiming initialization in deep learning. We leverage CoT initialization (**Sec 2.3**) for this exact reason: it provides a high-quality starting trajectory for our latent space search, which is **more effective than starting from random noise**.
> > > > 2. **Quantitative Improvement:** Our main results in Table 1 quantify this "improving power." The consistent, positive delta between our method and the baseline CoT (which serves as our initialization) demonstrates that our method is not merely preserving the initial performance but **actively improving upon it**. Even when the initial CoT isn't powerful enough, we can obtain a great improvement, such as an 16.22-point improvement over CoT when using Qwen2.5-1.5B-Instruct on the GSM8K dataset. Please refer to below for performance improvement table and case study.
> > > >
> > > > | Dataset | Prompt | Qwen2 7B | Qwen2.5 1.5B | Qwen2.5 7B | Qwen2.5 14B | LLaMA3.1 8B | Avg |
> > > > | :--- | :--- | :--- | :--- | :--- | :--- | :--- | :--- |
> > > > | **GSM8K** | Prompt 1 | +16.53 | +2.05 | +1.66 | +0.28 | +26.38 | +9.38 |
> > > > | | Prompt 2 | +12.05 | +16.22 | +12.43 | +0.76 | +7.48 | +9.79 |
> > > >
> > > >
> > > >
> > > >
> > > >
> > > > 3. **Qualitative Correction Example**: Quantitatively, there are many cases in the test set where the initial CoT generation fails, but LatentSeek **successfully corrects the trajectory** to find the right answer. We provide several case studies in **Appendix I**. And we show a clear example below:
> > > >
> > > >     ```
> > > >     QUESTION: For his 30th birthday, Elvira chose a new computer with many
> > > >     accessories as a gift. She has a budget of €1500 donated by her whole family
> > > >     and thinks that she will be able to keep a little money to afford a garment.
> > > >     She goes to a computer store and chooses a machine that costs €1090 with a screen,
> > > >     keyboard and mouse. She also takes a scanner for €157, a CD burner worth €74
> > > >     and a printer for €102. How much money will she have left for her clothing?
> > > >
> > > >     ---
> > > >
> > > >     TRUE ANSWER: 77
> > > >
> > > >     ---
> > > >
> > > >     ORIGINAL CoT (Incorrect):
> > > >     To determine [...], we need to follow these steps:
> > > >
> > > >     1. **Calculate the total cost of the computer and accessories:**
> > > >     [...]
> > > >     Therefore, the amount of money [...] is \boxed{0}.
> > > >
> > > >     ---
> > > >
> > > >     LatentSeek (Correct):
> > > >     Let determine [...] let need to calculate these steps:
> > > >
> > > >     1. CalculateIdent the total cost of all accessories accessories its chosen**
> > > >     [...]
> > > >
> > > >     2. Add the costs of the computer and accessories:
> > > >        \[
> > > >        1090 + 157 + 74 + 102 = 1423
> > > >        \]
> > > >
> > > >     3. Subtract the total cost from the budget:
> > > >        \[
> > > >        1500 - 1423 = 77
> > > >        \]
> > > >
> > > >     Therefore, [...] is \(\boxed{77}\).
> > > >     ```
> > > >
> > > > 4. **Compare with Self-Reflection**: The table below shows that LatentSeek **consistently outperforms** Self-Reflection [1], which is **an iterative test-time method that start with the same initial generation.** We report the average number across all 5 backbones and 2 prompts, indicating that LatentSeek holds a better optimization process:
> > > >
> > > >     | Dataset | GSM8K | MATH-500 |AIME2024 |
> > > >     | :--- | :--- | :--- |:---|
> > > >     | Self-Reflection | $70.65$ | $53.56$ | $4.00$|
> > > >     | LatentSeek | $80.24_{+9.59}$| $57.80_{+4.24}$ |$8.67_{+4.67}$ |
> > > >
> > > > [1] Reflexion: Language agents with verbal reinforcement learning.
> > > >
> > > > ---

---

> > > > > ### Author Response · Authors · 2025-11-20
> > > > > **Response to Reviewer EsPa (7)**
> > > > >
> > > > > > 7. In most reasoning sequences, errors tend to occur in the middle or later stages rather than at the beginning. Therefore, optimizing only the early latent tokens may not effectively address the actual sources of error. Overall, the method seems to perform slight adjustments or perturbations to the initial reasoning direction. I remain skeptical that its effectiveness arises mainly from random perturbations to the initial direction, rather than from genuine reinforcement learning with stable and informative reward signals—especially given that the reward employed in the paper appears highly noisy.
> > > > >
> > > > >
> > > > > We thank the reviewer for this insightful comment. The intuition behind  optimization of the first few tokens is that, **due to the autoregressive architecture, early tokens exert larger influence on the entire sequence**. As shown in Equation 4, although the source of error may originate from the middle or later stages, these later tokens are still **indirectly shaped by the early ones through the autoregressive dependency**.
> > > > >
> > > > > To further clarify this point, we include a comparison with **LatentSeek (middle)**, which instead optimizes **middle-stage tokens**, specifically, those occurring after 40% of the sequence length.
> > > > >
> > > > >
> > > > > | | Qwen2 | Qwen2.5 | Qwen2.5 | Qwen2.5 | LLaMA3.1 | Average |
> > > > > | :--- | :---: | :---: | :---: | :---: | :---: | :---: |
> > > > > |  | **7B** | **1.5B** | **7B** | **14B** | **8B** | **-** |
> > > > > | **GSM8K Prompt1** | | | | | | |
> > > > > | CoT | 68.01 | 68.08 | 88.86 | 92.03 | 50.19 | 73.43 |
> > > > > | LatentSeek (middle) | 74.68 | 69.07 | 89.31 | 92.49 | 65.21 | 78.15 |
> > > > > | LatentSeek (start) | **84.38** | **70.89** | **90.14** | **92.49** | **78.54** | **83.29** |
> > > > > | **GSM8K Prompt2** | | | | | | |
> > > > > | CoT | 65.20 | 15.31 | 66.41 | 91.81 | 69.07 | 61.56 |
> > > > > | LatentSeek (middle) | 70.73 | 22.44 | 72.93 | 92.49 | 77.26 | 67.17 |
> > > > > | LatentSeek (start) | **80.21** | **44.20** | **85.06** | **92.72** | **83.70** | **77.18** |
> > > > >
> > > > > We've added the experiments above and detailed discussions to **Appendix E.7 and Table 13**.
> > > > >
> > > > > ---

---

> > > > > > ### Author Response · Authors · 2025-11-20
> > > > > > **Response to Reviewer EsPa (8)**
> > > > > >
> > > > > > ### Questions:
> > > > > >
> > > > > >
> > > > > >
> > > > > > > Q1. I prefer the style and insight of the paper, but there are some issues that lead me to have some reservations. I lean more towards borderline (but there is no option for a score of 5). If the authors can address some of my concerns, I would be willing to raise my rating to acceptance.
> > > > > >
> > > > > > We are grateful for the reviewer’s kindness and insightful comments. Please let us know if there are any further questions or suggestions.

---

> > > > > > > ### Comment · Reviewer_EsPa · 2025-11-26
> > > > > > >
> > > > > > > Thanks to the author for his efforts in answering my questions, I will improve my score and wish everything the best~

---

> > > > > > > > ### Author Response · Authors · 2025-11-26
> > > > > > > >
> > > > > > > > Thank you for your time and for the positive update. Wishing you all the best as well!

---

### Official Review · Reviewer_G8rH · 2025-10-30

**Soundness:** 3
**Presentation:** 3
**Contribution:** 3
**Rating:** 6
**Confidence:** 4

**Summary:**

This paper introduces LatentSeek, a framework aimed at enhancing the reasoning capabilities of large language models (LLMs) by leveraging Test-Time Instance-level Policy Gradient optimization in the latent space. Unlike standard approaches that rely solely on explicit, step-by-step natural-language reasoning traces, LatentSeek iteratively refines the model’s latent representations based on a self-generated reward signal, allowing dynamic, per-instance adaptation of the reasoning trajectory. Empirical evaluations on multiple benchmarks (GSM8K, MATH-500, AIME2024) and across various LLM families (e.g., LLaMA, Qwen) show that LatentSeek outperforms established baselines, including Chain-of-Thought (CoT), Best-of-N (BoN), and training-based methods. Further analyses indicate that LatentSeek is computationally efficient, typically converging within a few iterations, and that performance improves with more latent update iterations. Overall, the study presents LatentSeek as a lightweight and effective paradigm for improving LLM reasoning without modifying model parameters.

**Strengths:**

1. This paper is highly novel, and its motivation is very clearly articulated.
2. The experiments in this paper are comprehensive, covering various large language models and multiple benchmarks. The analyses are thorough and convincingly demonstrate the effectiveness of the proposed method.
3. The methodology section is very clearly written, and the structured, formal presentation further clarifies how the method is applied.

**Weaknesses:**

1. This method is a instance-level policy gradient method. Each input instance requires separate iterative optimization of its latent representations, which reduces inference efficiency, especially for large-scale or complex tasks.
2. The updates occur in the latent space, making the internal changes of the model difficult to interpret. As a result, users or researchers may find it challenging to analyze or debug the model's reasoning process, reducing its practical usability.

**Questions:**

N/A

---

> ### Author Response · Authors · 2025-11-20
> **Response to Reviewer G8rH**
>
> We thank the reviewer for the kindness and the insightful review!
>
> > W1. This method is a instance-level policy gradient method. Each input instance requires separate iterative optimization of its latent representations, which reduces inference efficiency, especially for large-scale or complex tasks.
>
>
>
> We thank the reviewer for raising this important point about inference efficiency! The reviewer is correct that LatentSeek performs iterative optimization. We would like to clarify several points regarding our method's practical efficiency and scalability.
>
> 1. **The Per-Iteration Latent Optimization Cost is Negligible:** Importantly, the latent-space optimization updates for $z$ **only require the LM head**, rather than the full transformer backbone (as shown in Eq. 7). Each additional optimization step consists of **one backward pass and one forward pass through the LM head**. Based on our computation, a single optimization step incurs **only $\approx 0.69$ \%** of the FLOPs of a standard full forward pass. This additional cost is extremely small.
> 2.  **The Method Converges Quickly:** This small $\approx 0.69$ \% cost is incurred for very few steps. As shown in our Algorithmic Statistics (Table 9), LatentSeek converges rapidly, requiring **"fewer than two update iterations"** on average for GSM8K and MATH-500.
> 3. **Time Cost Comparison With Self-Reflection:** We compare LatentSeek with Self-Reflection, another iterative test time method. Evaluated on MATH-500, a relatively **complex tasks bench**, using Qwen2.5-7B-Instruct as backbone on GPU 4090, LatentSeek maintains **slightly less time cost** to self-reflection (**71.44s vs. 74.72s per instance**). Furthermore, LatentSeek holds stronger performance. In particular, it achieves an average of **4.24-point improvement over self-reflection on MATH-500**.
>
> 4. **Our Method is Parallelizable Across Instance**: As the reviewer notes, our method is instance-level. This is an advantage for large-scale tasks. The latent search for each instance is completely independent, which allows for trivial and massive parallelization across a batch of instances.
>
> Note:
> (1) Additional FLOPs calculation:
> * From Appendix G (Table 27), the FLOPs for the LM Head (decode stage) are $\approx 525\times 10^6 \text{ FLOPs}$.
> * Therefore, an estimate for one optimization step  is:
> $$
> Cost_{iter} \approx 3 \times Cost(\text{LM Head Forward Decode})
> $$
>
> $$
> \approx (1.575 \times 10^9) \ FLOPs
> $$
>
> $$
> (1.575 \times 10^9) / (2.29 \times 10^{11}) \approx 0.0069
> $$
>
> ---

---

> > ### Author Response · Authors · 2025-11-20
> > **Response to Reviewer G8rH (2)**
> >
> > > W2. The updates occur in the latent space, making the internal changes of the model difficult to interpret. As a result, users or researchers may find it challenging to analyze or debug the model's reasoning process, reducing its practical usability.
> >
> >
> > We thank the reviewer for this insightful comment! We agree that interpretability is a **common challenge for latent reasoning methods [1, 2]**, and our method also shares this challenge. We believe there is a **trade-off** in this research area between **maximizing reasoning effectiveness and maintaining full human-readable interpretability**.
> >
> > In our method, the optimized latents (e.g. $\rho = 0.2$, corresponding to the initial 20% of the thought process) serves as a *steering launch pad* for answer generation. The remaining tokens then leverage the base model’s strong autoregressive capability, yet still preserve **signs of logic**. This structure enables our method to maintain **a certain degree of interpretability**.
> >
> >
> > We provide several cases in **Appendix I**, and we'd like to show an example below:
> >
> > ```
> > QUESTION: For his 30th birthday, Elvira chose a new computer with many
> > accessories as a gift. She has a budget of €1500 donated by her whole family
> > and thinks that she will be able to keep a little money to afford a garment.
> > She goes to a computer store and chooses a machine that costs €1090 with a screen,
> > keyboard and mouse. She also takes a scanner for €157, a CD burner worth €74
> > and a printer for €102. How much money will she have left for her clothing?
> >
> > ---
> >
> > TRUE ANSWER: 77
> >
> > ---
> >
> > ORIGINAL CoT (Incorrect):
> > To determine [...], we need to follow these steps:
> >
> > 1. **Calculate the total cost of the computer and accessories:**
> > [...]
> > Therefore, the amount of money [...] is \boxed{0}.
> >
> > ---
> >
> > LatentSeek (Correct):
> > Let determine [...] let need to calculate these steps:
> >
> > 1. CalculateIdent the total cost of all accessories accessories its chosen**
> > [...]
> >
> > 2. Add the costs of the computer and accessories:
> >    \[
> >    1090 + 157 + 74 + 102 = 1423
> >    \]
> >
> > 3. Subtract the total cost from the budget:
> >    \[
> >    1500 - 1423 = 77
> >    \]
> >
> > Therefore, [...] is \(\boxed{77}\).
> > ```
> >
> > As this case demonstrates, the very beginning of our model's thought process ($\texttt{"...Let determine..., let need to calculate..."}$) is uninterpretable. This is the part decoded from the optimized latents.
> >
> > However, the subsequent reasoning process is **logical and human readable** (eg. $\texttt{"1090 + 157 + 74 + 102 = 1423"}$).
> >
> > This case shows that our method is not a complete "black box." We can still analyze the model's reasoning process to understand how it reached the correct answer.
> >
> > [1] Deng et al. 2024. From explicit cot to implicit cot: Learning to internalize cot step by step
> >
> > [2] Hao et al. 2024. Training Large Language Models to Reason in a Continuous Latent Space
> >
> > ---

---

### Official Review · Reviewer_UKFa · 2025-10-31

**Soundness:** 3
**Presentation:** 3
**Contribution:** 4
**Rating:** 8
**Confidence:** 4

**Summary:**

This paper proposes a novel framework, LATENTSEEK, designed to enhance the reasoning capabilities of large language models (LLMs) by applying instance-level policy gradient optimization in the latent space during inference. The motivation behind this approach is that, similar to human cognition, the optimal reasoning path for LLMs may not be limited to explicit natural language expressions but could instead leverage implicit, non-linguistic processes. The core idea of LATENTSEEK is to fix the model parameters during inference and iteratively optimize the latent representations of the input sequence to maximize a self-generated reward signal. This method bypasses the expensive cost of model fine-tuning. Experimental results across several reasoning benchmarks, including GSM8K, MATH-500, and AIME2024, as well as across different model families, show that LATENTSEEK significantly outperforms methods such as CoT, Best-of-N, and training-driven approaches.

**Strengths:**

1.The approach demonstrates significant innovation by employing policy gradient methods (e.g., REINFORCE) for instance-level optimization of continuous latent space representations during test-time inference to guide LLM reasoning.
2.The experimental results are highly promising, consistently achieving SOTA performance across multiple challenging mathematical reasoning datasets.
3.The framework exhibits high computational efficiency, typically converging within a few optimization iterations for tasks of moderate difficulty.

**Weaknesses:**

1.The reliability of the current self-generated reward function is questionable when faced with "incoherent" intermediate reasoning steps in the model's output. If the reward function is not sufficiently accurate, it may guide the latent space optimization towards local optima or incorrect reasoning paths.
2.In the derivation of the policy gradient, the authors assume that the latent representations $z$ are independent of each other. Is this assumption appropriate for a Transformer-based sequence model?

**Questions:**

See weakness.

---

> ### Author Response · Authors · 2025-11-20
> **Response to Reviewer UKFa**
>
> Thank you for your detailed and kind review!
>
> > W1. The reliability of the current self-generated reward function is questionable when faced with "incoherent" intermediate reasoning steps in the model's output. If the reward function is not sufficiently accurate, it may guide the latent space optimization towards local optima or incorrect reasoning paths.
>
>
>
> We thank the reviewer for the insightful comment. We discuss the need for better reward models as a key future direction in **Appendix A**.
>
> - **Reliable Self Reward:** We'd like to clarify that although our self-reward signal is not perfect, it provides a consistently **useful gradient for optimization**. Recent research has also demonstrated that large language models can serve as reliable reward sources [1, 2].
> - **Self Reward Accuracy:** To further *assess its reliability*, we conduct an *additional analysis on the GSM8K test set* in which the generation uses prompt 2 (the JSON prompt, defined in the paper). Specifically, we collect **all explored sequences**, including intermediate reasoning steps as well as final generated answers, and evaluate the alignment between the reward and ground-truth correctness. We binarized our dense [-1, 0] reward to "Positive" (Reward > -0.2) and "Negative" (Reward $\leq$ -0.2) and compared this against the ground-truth correctness. This yields the following results:
>
> |Model |Qwen2-7B-Instruct|Qwen2.5-1.5B-Instruct | Qwen2.5-7B-Instruct | Qwen2.5-14B-Instruct | Llama3.1-8B-Instruct| Average |
> | :---: | :---: | :---: | :---: | :---: |:---:| ---|
> | Acc | 85.78\% |77.15\% | 80.17\%|89.83\% | 80.46\%|82.68\%|
>
> As shown above, in $\sim 80\%$ cases, the self reward mechanism is reliable. In practice, the ability to judge an answer **cooperates with** the ability to generate one, and together they lead to substantially improved overall performance.
>
> - **Plot Trajectory:** To further illustrate the effect of the self-reward mechanism, we plot **the trajectory of the average reward over optimization steps**, as shown in the newly updated **Figure 5 (Appendix E.6)** of the manuscript. These plots clearly show that **the mean reward consistently increases across all models as the number of optimization steps increases**. It demonstrates that our reward signal **directionally guides the optimization process toward higher-reward regions**.
>
>
> We've added this discussion of our self-reward mechanism to **Sec 3.6** and **Appendix E.6**.
>
> [1] Yuan et al. 2024. Self-Rewarding Language Models.
>
> [2] Lifshitz et al. 2025. Multi-agent verification: Scaling test-time compute with multiple verifiers.
>
> ---

---

> > ### Author Response · Authors · 2025-11-20
> > **Response to Reviewer UKFa (2)**
> >
> > > W2. In the derivation of the policy gradient, the authors assume that the latent representations $z$ are independent of each other. Is this assumption appropriate for a Transformer-based sequence model?
> >
> >
> > Thanks for this insightful question! Our formulation for the derivation of the polict gradient, indeed assumes the independence of $z$. This is indeed an **empirical assumption** and we find it **empirically effective**. We'd like to make a further clarification as follows:
> >
> >
> > 1. **A Practical Design Choice:** Without this assumption, the autoregressive structure would *restrict optimization to only the first latent token*, since subsequent tokens depend on it, drastically shrinking the search space. By contrast, our independence assumption (optimizing multiple tokens in parallel) empirically yields **a larger search space**, allowing the sequence to optimize to a better place.
> > 3. **Empirical Validation:** The **good performance of LatentSeek (Table 1, 2, 3)** may also demonstrate that the assumption is a reasonable empirical choice.
> > 3. **A Steering Launch Pad:** The early decoded tokens serve as a launch pad that guides the model toward more effective reasoning trajectories. LatentSeek leverages the base model’s strong autoregressive capability: these initial tokens provide a flexible initialization for answer generation, and the independence assumption **brings more flexibility to the launch pad**.
> > 4. **Theoretical Guidance:** By drawing an analogy to Multi-Prover Interactive Proofs (MIP), in Appendix C, our analysis demonstrates that our framework's theoretical expressivity is strong enough, suggesting that the deficit of the independence assumption can be remedied by the base model's ability in reasoning tasks.
> >
> > We’ve added the explanation to the **Sec 2.2** in main text to explain more on the independence assumption.
> >
> >
> > ---

---

### Official Review · Reviewer_7PJX · 2025-11-04

**Soundness:** 3
**Presentation:** 3
**Contribution:** 3
**Rating:** 6
**Confidence:** 3

**Summary:**

The paper proposes LATENTSEEK, a training-free, test-time instance-level policy-gradient method that optimizes latent representations to improve reasoning, instead of optimizing prompts or model parameters. It initializes latents from a CoT decode, then iteratively updates a prefix of latents via REINFORCE using a self-generated reward, re-decoding after each update. Experiments on GSM8K, MATH-500, and AIME2024 with backbone models report consistent gains over CoT/BoN and several RL baselines.

**Strengths:**

1. The paper is generally well-written and structured, followed by the motivation and explanation.
2. The paper gives a compact REINFORCE formulation for test-time latent updates with token-factorization and per-latent gradients.
3. Experiments show average gains over several baselines and robust improvements on AIME2024.

**Weaknesses:**

1. The algorithm recursively computes the latent representations $z$ and decodes them back to $x$, which is more time-consuming compared to traditional latent-space methods that encode the text into latent representations only once. I suggest that the authors report the time or computational cost.
2. In Line 150, Equations (6) and (7), the authors assume independence among latent representations. However, transformer hidden states are typically highly correlated across time steps.
3. The hyperparameter $\rho$ appears to be crucial to model performance. It would be helpful for the authors to clarify how it is selected and whether any hyperparameter tuning was conducted.

**Questions:**

See Weaknesses

---

> ### Author Response · Authors · 2025-11-20
> **Response to Reviewer 7PJX**
>
> We thank the reviewer for the insightful comments! We'd like to offer further explanations as follows.
>
> > W1. The algorithm recursively computes the latent representations $z$ and decodes them back to $x$, which is more time-consuming compared to traditional latent-space methods that encode the text into latent representations only once. I suggest that the authors report the time or computational cost.
>
>
>
> Thanks for the suggestion!
>
> 1. We first emphasize that after the CoT-based initialization, the initial latent representations $z$ are updated using gradients flowing only through the language-model head. Tokens are then decoded via a forward pass through the same head. In this way, **LatentSeek avoids full forward-backward passes through the entire model**, making the procedure lightweight in terms of computation.
>
>
> 2. Second, we present a FLOPs comparison against a representative training-based latent-space method, iCoT [1], in the table below. iCoT is a representative of this class of approaches, and is similar to other methods in the category. The result is reported using Llama3.1-8B-Instruct evaluated on the GSM8K test set (1319 instances).
>
>
>
>
> |            | Type   | #Train Dataset | #Train Epoch | #Test Dataset | Acc (%) (GSM8K $\uparrow$) | Total FLOPs ($\downarrow$)             |
> | ---------- | ------ | -------------- | ------------ | ------------- | --------------- | ----------------------- |
> | LatentSeek | Latent | 0              | 0            | 1.3K          | **83.7**        | $7.69 \times 10^{14}$|
> | iCoT       | Latent | 37.8K          | 200          | 1.3K          | 47.08           | $5.19 \times 10^{18}$ |
>
>
> As shown in the table above, the **total FLOPs of LatentSeek are roughly $10^4$ times smaller than those of iCoT**. This efficiency comes from **eliminating the need of training**: LatentSeek performs latent updates *only at test time*, and each update requires just *a single "forward–backward" pass of the language-model head*, making the procedure efficient. Moreover, as shown in **Table 9,** LatentSeek typically converges in fewer than two update iterations during inference.
>
> *We've added the FLOPs calculation and comparison to Appendix G.4.*
>
>
> [1] Deng et al. 2024. From Explicit CoT to Implicit CoT: Learning to Internalize CoT Step by Step
>
>
>
>
>
>
>
>
>
>
> ---

---

> ### Author Response · Authors · 2025-11-20
> **Response to Reviewer 7PJX (2)**
>
> > W2. In Line 150, Equations (6) and (7), the authors assume independence among latent representations. However, transformer hidden states are typically highly correlated across time steps.
>
>
> Thank you for this insightful question. You are correct that transformer's hidden states are typically correlated, and our formulation, Equations (6) and (7), makes a simplifying assumption of independence for the gradient update. This is indeed an **empirical assumption** and we find it **empirically effective**. We'd like to clarify the rationality of the assumption as follows:
>
>
> 1. **A Practical Design Choice:** Without this assumption, the autoregressive structure would *restrict optimization to only the first latent token*, since subsequent tokens depend on it, drastically shrinking the search space. By contrast, our independence assumption (optimizing multiple tokens in parallel) empirically yields **a larger search space**, allowing the sequence to optimize to a better place.
> 3. **Empirical Validation:** The **good performance of LatentSeek (Table 1, 2, 3)** may also demonstrate that the assumption is a reasonable empirical choice.
> 3. **A Steering Launch Pad:** The early decoded tokens serve as a launch pad that guides the model toward more effective reasoning trajectories. LatentSeek leverages the base model’s strong autoregressive capability: these initial tokens provide a flexible initialization for answer generation, and the independence assumption **brings more flexibility to the launch pad**.
> 4. **Theoretical Guidance:** By drawing an analogy to Multi-Prover Interactive Proofs (MIP), in Appendix C, our analysis demonstrates that our framework's theoretical expressivity is strong enough, suggesting that the deficit of the independence assumption can be remedied by the base model's ability in reasoning tasks.
>
> We’ve added the explanation to the **Sec 2.2** in main text to explain more on the independence assumption.
>
>
> ---

---

> > ### Author Response · Authors · 2025-11-20
> > **Response to Reviewer 7PJX (3)**
> >
> > > W3. The hyperparameter $\rho$ appears to be crucial to model performance. It would be helpful for the authors to clarify how it is selected and whether any hyperparameter tuning was conducted.
> >
> >
> >
> > We thank the reviewer for pointing out! The fraction ratio, $\rho$, is indeed an important hyperparameter in our method.
> >
> > We did conduct hyperparameter tuning to select $\rho$. Specifically, for each model, we *performed a hyperparameter search over a small, randomly selected validation set from the original dataset*, such as for GSM8K, we select a random sample of 500 data from the GSM8K training set (See below for an example of tuning Qwen2.5-7B-Instruct). We selected the $\rho$ value that **yielded the best performance on this small validation set** for our main experiments.
> >
> > | $\rho$   | 0.1   | 0.2   | 0.4   | 0.6   | 0.8   |
> > | --- | ----- | ----- | ----- | ----- | ----- |
> > | GSM8K Val Acc     | 90.40 | **91.00** | 85.60 | 82.00 | 81.20 |
> >
> > We've added this above to **Appendix F.4** clarifying this tuning process.
> >
> > ---

---

### Author Response · Authors · 2025-11-29
**Highlight of Contributions and Summary of Rebuttal**

Dear AC,

Thank you for your time! We appreciate the reviewers’ constructive feedback. In response, we have conducted additional experiments and analyses, revising the manuscript accordingly. All changes are *highlighted in red*. Please find a summary of the reviews and our responses below.
## Highlights of Contributions
1. **High Novelty:** Our method, a test-time instance-level policy gradient approach, *"is highly novel"* (**Reviewer G8rH**), *"interesting"* (**Reviewer EsPa**) and *"demonstrates significant innovation"* (**Reviewer UKFa**), with our motivation *"very clearly articulated"* (**Reviewer G8rH**). What's more, our approach *" bridges reinforcement-based adaptation with test-time inference"* (**Reviewer EsPa**).
2. **Robust Evaluation and High Performance:** By **Reviewer G8rH**, the experiments in our paper are *"comprehensive, covering various large language models and multiple benchmarks"* and our analyses are *"thorough and convincingly demonstrate the effectiveness of the proposed method"*.  Our experimental results are *"highly promising"* (**Reviewer UKFa, EsPa**). Our method shows *"average gains over several baselines and robust improvements on AIME2024"* (**Reviewer 7PJX**) and *"outperforms established baselines, including Chain-of-Thought (CoT), Best-of-N (BoN), and training-based methods"* (**Reviewer G8rH**), *"consistently achieving SOTA performance across multiple challenging mathematical reasoning datasets"* (**Reviewer UKFa**).
3. **Insightful Analysis:** **Reviewer G8rH** comments that our analyses are *"thorough and convincingly demonstrate the effectiveness of the proposed method"*. Furthermore, **Reviewer EsPa** finds the concept of *"improving the quality of CoT paths by optimizing the latent space"* to be *"interesting"* and confirms that our work *" provides insight into test-time scaling"*, and he/she *"prefers the style and insight"* of our paper.
4. **High Efficiency:** Our framework *"exhibits high computational efficiency"* (**Reviewer UKFa**), *"typically converging within a few optimization iterations"* (**Reviewer UKFa, EsPa**).
## Summary of Rebuttals
1. **Further Clarification on the Independence Design Choice (Reviewers 7PJX, UKFa, EsPa):** We have expanded our discussion of the independent-update strategy in *Section 2.2*. In particular, we provide additional justification for this **empirical design choice**, explaining its rationale from both **theoretical motivations** and **empirical considerations related to search-space expansion**.
2. **Additional Ablation Analysis:**
    * **Comparison with Unguided Search (Reviewer EsPa):** We compare LatentSeek with a single-step stochastic explorative search in latent space (*Sec 3.5*), confirming that our **performance gains arise from gradient guidance rather than random perturbations**.
    * **Analysis of the Self-Reward Mechanism (Reviewer EsPa, UKFa):** We perform additional experiments comparing with constant reward (*Sec 3.6*), list the accuracy of the self reward mechanism (*Sec 3.6*) as well as plot the trajectory of mean rewards (*Fig 5. Appendix E.6*), demonstrating the effectiveness of the self-reward mechanism.
    * **Middle-Stage Optimization (Reviewer EsPa):** We included experiments in *Appendix E.7* comparing initial-stage versus middle-stage optimization, demontrating the **effectiveness of the previous one**.
3. **Mechanism Explanation:**
    * **Latent Dynamics (Reviewer EsPa):** We analyzed the dynamics of the latent representations, by 1) adding visualizations of Latent Trajectories (**Appendix H**), 2) demonstrating the macro-level movement toward high-reward regions (**Appendix E.6**), 3) explaining the gradient directions, and 4) mentioning already written observations on semantic drift (**Appendix I, Figure 11**).
    * **Interpretability (Reviewer G8rH):** We discussed the trade-off between effectiveness and interpretability. As demonstrated by the case studies in **Appendix I,** while the optimized tokens may be incoherent, the subsequent reasoning process preserves **signs of logic**.
4. **Additional Efficiency Analysis:**
    * **Comparison with Other Latent-Space Methods (Reviewer 7PJX):** The FLOPs comparison in **Appendix G.4** against `iCoT`, a representative training-based latent method, shows that LatentSeek reduces total FLOPs by approximately **$10^4 \times$.**
    * **Inference Efficiency (Reviewer G8rH):** We analyzed the negligible per-iteration cost and the method's rapid convergence. We also included a time-cost comparison with Self-Reflection, demonstrating that LatentSeek achieves superior performance with comparable efficiency.

In summary, we believe we have comprehensively **addressed all concerns** raised by the reviewers, especially those from **Reviewer EsPa**. And we are encouraged by **Reviewer EsPa's** positive acknowledgement and **his/her decision to increase the score**. Thanks again for your time and your work!

Best regards,

The authors

---

### Note · Program_Chairs · 2026-01-17
**Submission Desk Rejected by Program Chairs**

The following references in this submission do not refer to real documents and/or have major errors in bibliographic information:

 Yao Qin, Asli Celikyilmaz, Wenjing Li, Jung H Oh, Victoria Lin, Semih Agrawal, Yang Zhou, Singaram Kumar, Jing Shen, Mitesh M Khapra, et al. Pomp: Unsupervised controllable generation by optimizing prompts via reinforcement learning. IEEE Transactions on Neural Networks and Learning Systems, 2023.
David Sun, Jasmine Jiang, Terry Nguyen, James Tsai, Yizhe Xia, Kyunghyun Cho, Heng Ji, Hugo Larochelle, Xiang Yuan, David Simchi-Levi, et al. Recitation-augmented language models. In International Conference on Learning Representations (ICLR), 2023.